# Judgments of agency are affected by sensory noise without recruiting metacognitive processing

**Marika Constant**[1,2,3]*, **Roy Salomon**[4], **Elisa Filevich**[1,2,3]

[1]Faculty of Life Sciences, Department of Psychology, Humboldt-Universität zu Berlin, Berlin, Germany; [2]Bernstein Center for Computational Neuroscience Berlin, Berlin, Germany; [3]Berlin School of Mind and Brain, Humboldt-Universität zu Berlin, Berlin, Germany; [4]Gonda Multidisciplinary Brain Research Center, Bar-Ilan University, Ramat-Gan, Israel

**Abstract** Acting in the world is accompanied by a sense of agency, or experience of control over our actions and their outcomes. As humans, we can report on this experience through judgments of agency. These judgments often occur under noisy conditions. We examined the computations underlying judgments of agency, in particular under the influence of sensory noise. Building on previous literature, we studied whether judgments of agency incorporate uncertainty in the same way that confidence judgments do, which would imply that the former share computational mechanisms with metacognitive judgments. In two tasks, participants rated agency, or confidence in a decision about their agency, over a virtual hand that tracked their movements, either synchronously or with a delay and either under high or low noise. We compared the predictions of two computational models to participants' ratings and found that agency ratings, unlike confidence, were best explained by a model involving no estimates of sensory noise. We propose that agency judgments reflect first-order measures of the internal signal, without involving metacognitive computations, challenging the assumed link between the two cognitive processes.

*For correspondence:
marika.constant@gmail.com

Competing interest: The authors declare that no competing interests exist.

## Editor's evaluation

This article describes a carefully designed study on the computational mechanisms underlying judgements of agency in an action-outcome delay task. Model-based analyses of behavior indicate that, unlike judgments of confidence, judgments of agency do not recruit metacognitive processes. This finding is important, because it challenges the assumed relation between agency and metacognition.

## Introduction

Attributing ourselves agency, or causation of our actions and their outcomes, is central to our experience of moving intentionally. Previous research has suggested that agency depends on a comparison between the predicted and observed consequences of our actions, resulting in a prediction error signal if they do not match. Under this widely accepted comparator model of agency, it is the prediction error signal that leads to our feeling of agency (FoA), which we assess in order to make judgments of agency (JoAs) (*Carruthers, 2012*; *Frith et al., 2000*; *Haggard, 2017*). There are several sources of noise to these signals, and agency processing, much like other perceptual processing, occurs under varying degrees of uncertainty. Noisy agency signals are suggested to play a role in clinical cases involving striking disruptions to agency, such as in schizophrenia (*Corlett et al., 2010*; *Fletcher and*

*Frith, 2009*; *Moore and Fletcher, 2012*; *Robinson et al., 2016*), so it is critical to understand precisely how agency signals change with uncertainty, in both healthy and clinical populations.

Some existing models of agency have suggested that noise affects the comparator signal because the latter is the result of a cue integration process, in which the motor and sensory information are integrated and weighted by their reliability (*Moore and Fletcher, 2012*; *Moore and Haggard, 2008*). In line with these models, participants' FoA, as measured by an implicit temporal binding effect, has been found to rely less on cues that are noisy or imprecise, and more on more informative cues (*Moore et al., 2009*; *Wolpe et al., 2013*). Further, one recent study put forward a possible computational model in which FoAs are affected by sensory noise (*Legaspi and Toyoizumi, 2019*). However, work on uncertainty in lower level agency processing has not addressed the role of noise in the formation of JoAs. It is not clear that explicit JoAs incorporate noise in the same way as lower level FoAs, as JoAs are generally considered to be at a higher level of the processing hierarchy (*Gallagher, 2007*; *Haggard and Tsakiris, 2009*; *Sato, 2009*; *Synofzik et al., 2008*).

Explicit JoAs could incorporate noise in two main ways. One alternative is that JoAs are simply reports of FoAs, that do not involve secondary assessments of their reliability. A second alternative is that JoAs monitor the uncertainty of the first-order signal through computations that include a secondary assessment of the sensory uncertainty. The crucial difference between these two alternatives is whether JoAs require an estimate of the precision of the first-order signal, or can be determined without any information about it. This kind of second-order representation of the precision of a first-order signal is a defining feature of what has been called a metacognitive process (*Shea, 2012*). Accordingly, metacognitive judgments are often operationalized as confidence ratings following discrimination responses (*Fleming and Lau, 2014*), and these second-order metacognitive ratings have been shown to monitor the uncertainty of the perceived internal signal (*de Gardelle et al., 2015*; *Navajas et al., 2017*; *Rausch et al., 2017*; *Sanders et al., 2016*; *Spence et al., 2016*).

In the literature, JoAs have sometimes been considered to be metacognitive reports about otherwise first-order agency signals (*Carruthers, 2015*; *Metcalfe et al., 2012*; *Metcalfe and Greene, 2007*; *Miele et al., 2011*; *Potts and Carlson, 2019*; *Terhune and Hedman, 2017*; *Wenke et al., 2010*; *Zalla et al., 2015*). However, it is unclear if the link to metacognition is merely conceptual, or whether JoAs result from second-order uncertainty representations, and are computationally comparable to other metacognitive judgments. Here, we commit to a definition of metacognition as a process that involves second-order uncertainty monitoring computations, to test the link between JoAs and metacognition in this narrow but computationally tractable sense.

We propose that for agency judgments to be metacognitive in a computational sense, they should monitor the noise in the perceived prediction error signal, incorporating a second-order judgment of the uncertainty of one's agency processing. We examined this by setting up a two-criterion test. The two criteria refer to behavioral and computational measures, respectively. The first criterion for JoAs to be computationally metacognitive was for sensory noise to influence JoAs in a secondary way, beyond just altering their variance across trials. We expected the effect of the delay to become smaller in high perceptual uncertainty trials, leading JoAs to be less extreme, as compared to trials with lower uncertainty. This would suggest that the reliability of the signal was estimated and factored in for each rating, against the simpler alternative that the rating was made on the basis of a linear readout of the less reliable signal. We examined this by assessing the effects of noise and delay on explicit agency ratings, using a sensory noise manipulation orthogonal to the delay.

This first criterion formed our pre-registered hypothesis and was necessary, but not sufficient, for JoAs to be considered metacognitive. It therefore served as a prerequisite for the second criterion: Agency and confidence judgments should show similar sensitivity to internal estimates of the sensory noise, suggesting the involvement of second-order uncertainty monitoring following the same computational principles. We assessed this by contrasting two computational models against distributions of JoA data, one that would reflect metacognitive monitoring of noise and an alternative model that would not. We found that JoAs satisfied the first, but not the second criterion: Sensory noise did indeed influence JoAs, but this influence did not reflect any second-order noise monitoring, suggesting that JoAs may not be metacognitive in the computational sense.

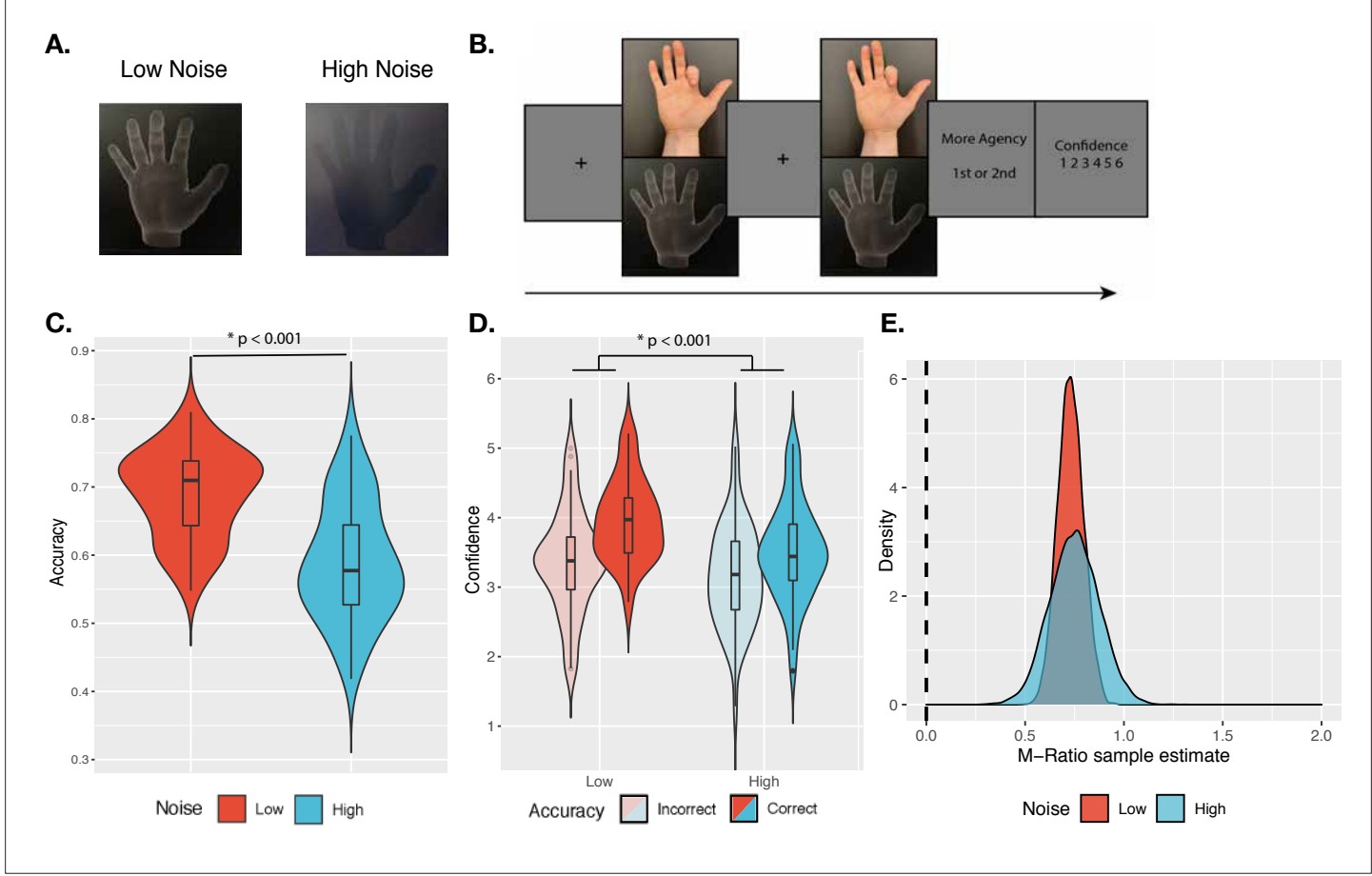

**Figure 1.** Confidence task. (**A**) Sensory noise conditions. Sensory noise was manipulated by changing the illumination, with high sensory noise captured by low contrast, dark illumination, and low sensory noise captured by high contrast, bright illumination. (**B**) Experimental paradigm. Cued by the offset of the fixation cross, participants made two consecutive finger movements on each trial, with their hand out of sight. On the screen, participants saw either a virtual hand moving in synchrony with their own, or with an additional temporal delay. Participants first discriminated which movement they felt more agency over, and then rated their confidence in their own response. (**C**) Discrimination accuracy. In line with the intended effect of the sensory noise manipulation, accuracy was significantly higher for low- vs. high-noise conditions. (**D**) Mean confidence ratings. We found a significant interaction effect between Response Accuracy and Noise on Confidence. Violin plots capture the kernel probability density and boxplots show the median, interquartile range (IQR) with hinges showing the first and third quartiles, and vertical whiskers stretching to most extreme data point within 1.5*IQR from the hinges. Outliers are plotted as black or gray dots. (**E**) M-Ratio Estimates. Metacognitive efficiency for each noise level, estimated using the HMetad' toolbox. The dashed vertical line indicates chance-level metacognitive efficiency.

## Results

Each participant completed two tasks: A confidence-rating task, consisting of a two-interval forced choice (2IFC) followed by a confidence rating on a scale from 1 to 6; and an agency-rating task consisting of a JoA on an equivalent scale. Both tasks used the same basic stimuli, namely, a movement of participants' index finger tracked by a LEAP Motion infrared tracker and displayed on the screen as a virtual hand movement either in synchrony or with small temporal delays. In both tasks, we manipulated sensory noise in the same way, by changing the illumination of the scene. We created two conditions (low and high sensory noise) by displaying the virtual hand under bright, high-contrast illumination or under dim, low-contrast illumination respectively (*Figure 1A*). This change in illumination, a manipulation that was orthogonal to the stimulus intensity, effectively changed the reliability of the mapping between the external and internal signal without affecting the stimulus.

**Table 1.** Syntax for the linear mixed-effects models used.

| Task | Hypothesis | Model formula |
| --- | --- | --- |
| Confidence Task | Sensory noise influences response accuracy | logit(Response Accuracy)~ Noise + (1 | Participant) |
| Confidence Task | Sensory noise influences confidence following correct decisions | Confidence~ Response + Response Accuracy*Noise + (1 | Participant) |
| Agency Rating Task | Sensory noise influences the effect of delay on JoA | JoA ~ Delay*Noise + (Delay:Noise | Participant) |

## Confirmatory analyses

### Confidence-rating task

On each trial of the confidence task, participants were cued to make two consecutive movements (which constituted the two intervals of the 2IFC) of their right index finger, with their hand out of sight. For only one of the two movements, we added a temporal delay to the virtual hand shown on the screen (in the other interval, the virtual hand was displayed to match the participant's hand movement in real time). Participants then discriminated in which interval they felt more agency over the movement of the virtual hand, and rated confidence in their response (*Figure 1B*). We assumed that participants compared their degree of control over the virtual hand in the two movements to solve the task, and rated confidence in this comparison. This paradigm brings agency into a standard framework for studying metacognition (*Wang et al., 2020*). Importantly, under this operationalization, we can define correct responses to the 2IFC discrimination task as those where participants report that they felt more agency for the stimulus without any added delay, allowing us to quantify discrimination accuracy. If the illumination manipulation served to increase sensory noise in the intended way, we expected lower discrimination accuracy under high sensory noise compared to low noise (*Macmillan and Creelman, 1991*). Further, based on previous work using similar confidence paradigms (*Bang and Fleming, 2018*; *de Gardelle et al., 2015*; *Spence et al., 2016*), and a normative model of confidence (*Sanders et al., 2016*), we predicted an interaction between sensory noise and accuracy on confidence, in particular with confidence decreasing in high noise following correct trials and increasing in high noise following incorrect trials. To test the effect of the illumination manipulation, we first built a logistic regression model on response accuracy, including sensory noise as a fixed effect, and by-participant random intercepts (see *Table 1* for the explicit model syntax). As expected, we found a main effect of Noise, revealing significantly lower accuracy in the high-noise compared to the low-noise condition (Mdiff = 10%, SE = 1.4%, $\chi^2(1)$ = 97.60, p < 0.001, $BF_{10}$ = 1.78 × $10^{20}$, OR = 1.55, 95% CI [1.42, 1.70]; *Figure 1C*).

Then, we built a linear mixed-effects model on confidence to test the second prediction (an interaction effect between sensory noise and response accuracy). The model included the interaction between response accuracy and noise level and each factor as fixed effects, as well as by-participant random intercepts. We also included response identity (first or second interval) as a fixed effect, as presentation order could have biased confidence ratings (*Jamieson and Petrusic, 1975*; *Yeshurun et al., 2008*). In line with our predictions, we found a significant interaction between Noise and Response Accuracy on confidence, F(1,8858) = 14.43, p < 0.001, $BF_{10}$ = 2.19, $\eta^2_p$ = 0.0016 (*Figure 1D*), with a stronger difference in confidence between correct and incorrect trials under low noise ($Mdiff_{Correct-Incorrect}$ = 0.58, SE = 0.044) compared to high noise ($Mdiff_{Correct-Incorrect}$ = 0.34, SE = 0.042). In addition to the interaction effect, we found that confidence following incorrect decisions was lower in the high-noise compared to the low-noise condition. Although we expected confidence following incorrect decisions to increase under high noise, the 'double-increase' confidence pattern seen here has also been shown in the literature (*Adler and Ma, 2018*; *Rausch et al., 2017*; *Rausch et al., 2019*). Finally, we found a significant main effect of Response, F(1,8872) = 82.64, p < 0.001, $BF_{10}$ = 5.71 × $10^{14}$, $\eta^2_p$ = 0.0092, with pairwise comparisons revealing significantly higher confidence ratings when participants reported feeling more agency over the stimulus in the second interval, compared to the first (Mdiff = –0.27, SE = 0.030), t(8872) = –9.09, p < 0.001. These results were also all confirmed by repeating this analysis with ordinal models (Appendix 2).

In order to further ensure that participants' confidence ratings reflected metacognitive processing, especially given relatively low accuracy under high noise, we checked that their metacognitive

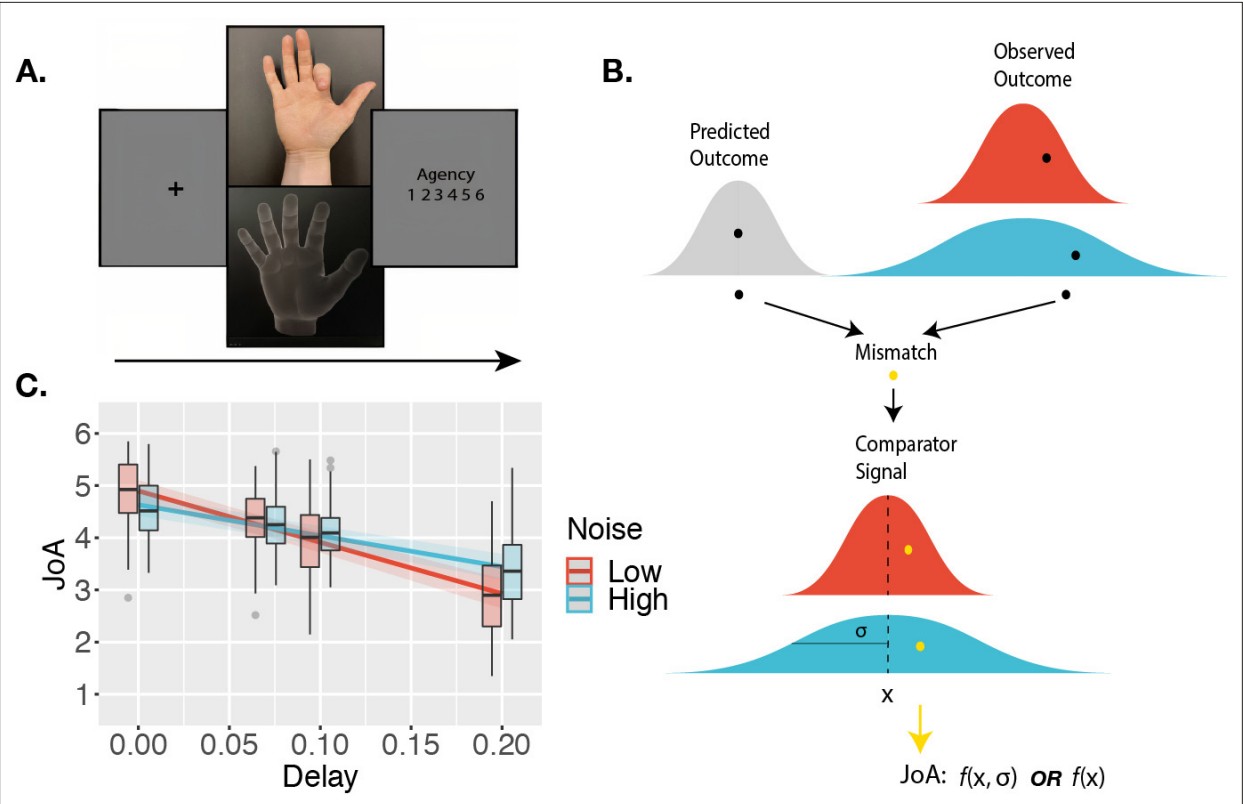

**Figure 2.** Agency-rating task. (**A**) Experimental paradigm. Cued by the offset of the fixation cross, participants made one finger movement on each trial, with their hand out of sight. On the screen, participants saw either a virtual hand moving in synchrony with their own, or with an additional delay/ temporal lag. Participants then made a judgment of the degree of agency they experienced. (**B**) Sketch of noise dependency prediction. By decreasing the illumination of the scene, the high-noise condition adds sensory noise to the observed outcome, and hence to the comparator signal, computed as the mismatch between predicted and observed outcomes. This environmental noise, independent of the delay, leads to a noisier mapping between the actual delay in the environment and the perceived internal signal. According to the comparator model, this internal delay signal is then compared to the representation of the predicted outcome, leading to the internal comparator signal. The internal comparator signal is represented as the bottom-most red (low uncertainty) and blue (high uncertainty) probability distributions. Points reflect the perceived signals on single trials. If the JoA is a readout of the comparator signal on each trial, though the variance would increase with noise, the mean JoA will not depend on noise, but will reflect the mean of the comparator signal distribution. Alternatively, if JoAs monitor the noisiness of the comparator signal, mean JoA will depend on noise. In other words, JoAs would be a function not only of a noisy signal (x), but of the noisy signal and the noise itself (σ). (**C**) Interaction effect result. Predicted JoA across delays and noise conditions from linear mixed-effects model results. 95% confidence intervals shown. Boxplots reflect subjectwise mean JoAs per noise level and delay. They show the median, interquartile range (IQR) with hinges showing the first and third quartiles, and vertical whiskers stretching to most extreme data point within 1.5*IQR from the hinges. Outliers are plotted as gray dots.

efficiency (M-Ratio) was above-chance in both conditions (*Figure 1E*; see Appendix 3 for full analysis). Together, these results suggest that the illumination manipulation affected sensory noise as we intended, and influenced metacognitive confidence judgments as previous studies have shown.

## Agency-rating task

On each trial of the agency-rating task, participants made a single movement of their index finger and watched the virtual hand model either move synchronously with their movement (25% of trials) or with a delay of either 70, 100, or 200 ms. After every movement, participants rated their agency on a scale from 1 to 6 (*Figure 2A*). By adding sensory noise to the perceived sensory outcome of the movement (namely, the virtual hand movement), we added noise to the comparator signal that participants assessed with their JoAs (*Figure 2B*). We then formulated a two-criterion test to assess whether JoAs are computationally metacognitive, and hence involve second-order estimates of the noise. The first — behavioral — criterion was that mean JoAs should depend on both delay and sensory noise. This would suggest that sensory noise influences agency ratings beyond just increasing their variability, with agency ratings instead including an indication of the signal's precision (*Figure 2B*). Alternatively,

if the mean JoA per delay does not depend on the sensory noise level, this would indicate that JoAs are simply a first-order report of the perceived comparator signal, with mean JoA reflecting the mean of the comparator signal distribution (*Figure 2B*). The second criterion of our test, if the first criterion was met, was at the computational level, for agency ratings to involve underlying metacognitive computations such as those involved in confidence, in particular, the second-order monitoring of sensory noise. To test this, we compared two computational models built based on the results of the first criterion, a Second-order agency model that involves metacognitive processing, and an alternative First-order model that does not.

## Behavioral results

While the results of the confidence task confirmed that the illumination manipulation affected sensory noise overall as intended, we were interested in examining precisely how JoAs responded to sensory noise, and hence required that the high-noise condition actually increased sensory noise for all participants included in the analysis of the agency rating task. We therefore excluded from the following analyses any participants for whom discrimination accuracy in the high-noise condition was not lower than in the low-noise condition in the confidence task. We note that all the results described below remain largely the same, and the conclusions unchanged, when we include the data from all participants in the analyses (see Appendix 1). We predicted that if sensory noise affected JoAs similarly to metacognitive processes, we would observe a significant interaction between Noise and Delay on JoA (*Figure 2B*). This would be the first of our two criteria. In particular, if uncertainty led to a metacognitive scaling of JoAs, similarly to confidence-weighted agency ratings, we would expect to see less extreme JoAs under high noise, with low certainty pulling ratings closer to the decision criterion. We investigated this using a linear mixed-effects model on JoAs that included the interaction between noise level and delay as fixed effects, and allowed for by-participant random effects of the interaction, and random intercepts (*Table 1*). We found a significant interaction effect between Noise and Delay, $F(1,52) = 61.16$, $p < 0.001$, $BF_{10} = 3.78 \times 10^6$, $\eta^2_p = 0.54$, 95% CI [0.35, 0.67], with a less extreme negative slope across delay values in the high-noise condition ($\beta_{High} = -5.93$, $SE = 0.69$), compared to low-noise ($\beta_{Low} = -9.84$, $SE = 0.77$) (*Figure 2C*), suggesting that JoAs met our first criterion. We also found a significant main effect of Delay, $F(1,39) = 132.05$, $p < 0.001$, $BF_{10} = 14486.52$, $\eta^2_p = 0.77$, 95% CI [0.64, 0.85], replicating previous findings that showed increasing delays of the virtual hand movement to lead to lower JoAs (*Krugwasser et al., 2019*; *Stern et al., 2020*). We found this effect of JoAs decreasing with delay for the majority of participants (37 out of 40) included in this analysis in both conditions, indicating that participants were able to make meaningful ratings, even in the high-noise condition. We also repeated these analyses with ordinal models, which confirmed our results (Appendix 2).

## Exploratory analyses
### Computational modeling

We found in our behavioral analysis that the mean JoA depended on both delay and noise, meeting our first criterion for JoAs being metacognitive. This allowed us to move on to our second test-criterion and investigate whether there are strictly metacognitive computations underlying agency judgments. In order to test our second criterion, namely, whether JoAs can be explained by the same computations as confidence, we compared two possible models of agency ratings (the Second-order model and the First-order model, *Figure 3A*), that differed in their predicted distributions of JoAs across noise conditions and delays (*Figure 3B*). Both models could in principle account for the observed interaction effect between noise and delay on JoA, satisfying the first criterion. However, only the Second-order model required participants to be able to reflect on the noise of their own sensory processing. The Second-order model assumed that agency ratings behave like confidence as described by Bayesian-confidence models, namely as the posterior probability that a decision is correct, given the strength of the internal evidence and the decision (*Navajas et al., 2017*; *Sanders et al., 2016*). The computation of this probability requires the observer to have second-order access to estimate their own sensory noise (*Figure 3A*), beyond just adopting precision-weighted cues from the lower level. So, in this model, JoAs are metacognitive because they require a representation of the precision of the first-order agency signal.

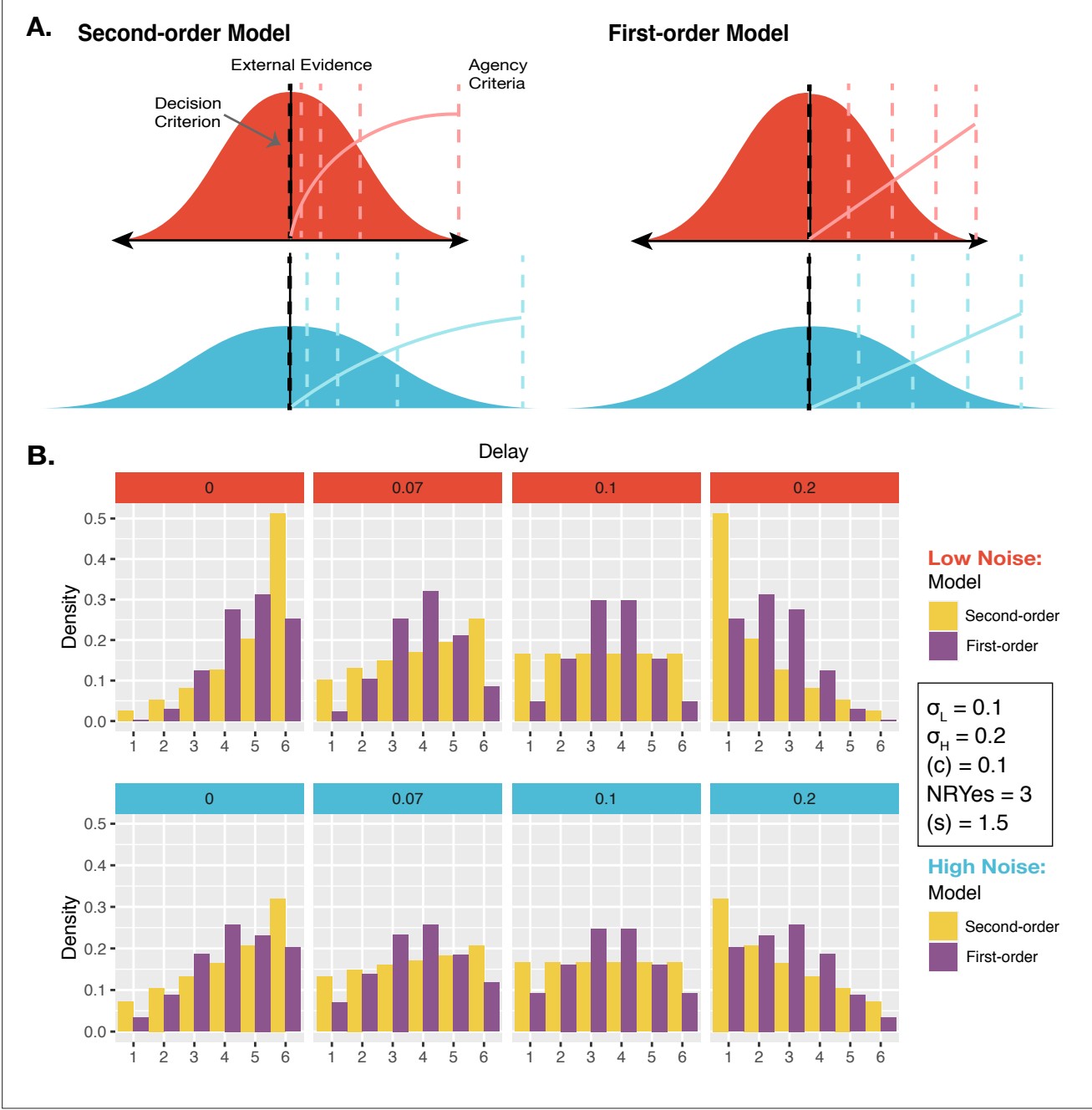

**Figure 3.** Models and Predictions. (**A**) Two models of JoAs. In the Second-order model of agency, JoA reflects the posterior probability of the detection decision being correct, given the choice and internal evidence. The Gaussian distributions are probability distributions of the internal comparator signal. The dashed colored lines indicate the criteria, or the thresholds on the internal signal axis that delineate the different JoA values. These are spaced linearly in probability space, so their positions on the internal signal space change with noise level. The solid colored lines show the expected JoAs as a function of internal signal strength. In the First-order model of agency, JoA reflects a first-order estimate of delay compared to the criterion, not based on noise. However, the criteria are spread evenly across the range of signals within each noise level, such that they interact with noise level only due to a rescaling of ratings. The solid colored lines show the predicted JoA as a function of internal signal strength. Note that this function is linear but still interacts with noise, due to the rescaling. (**B**) Example model predictions. To illustrate the predictions of each of the models, we chose representative parameter values to run simulations. These values fall within the range of the values resulting from the model fits to participants' data. Predicted distributions of JoAs per delay and noise condition are shown as densities.

As an alternative, we considered the First-order model, which parallels the Second-order one except that ratings are based on first-order point-estimates of evidence and therefore do not factor in metacognitive estimates of sensory noise (*Figure 3A*). This First-order model accounts for the observed relationship between noise and JoAs by considering that participants might simply treat the noise conditions independently. In practical terms, this implies that participants judged agency in low-noise trials relative to one another, and high-noise trials relative to one another. Although our design aimed to prevent this by interleaving the conditions, it is still possible that participants did this to some extent, as our noise manipulation was visually very apparent. Critically, this condition-dependent scaling is achieved without making any estimates of the sensory noise of the conditions. The only role noise plays in this model is to make the conditions distinguishable from one another such that participants might treat them independently and use the rating scale accordingly. This means that in each condition, the maximum of the scale, or a '6', refers to the maximum agency experienced only within trials of that condition. Therefore, in the First-order model, a '6' may differ between conditions

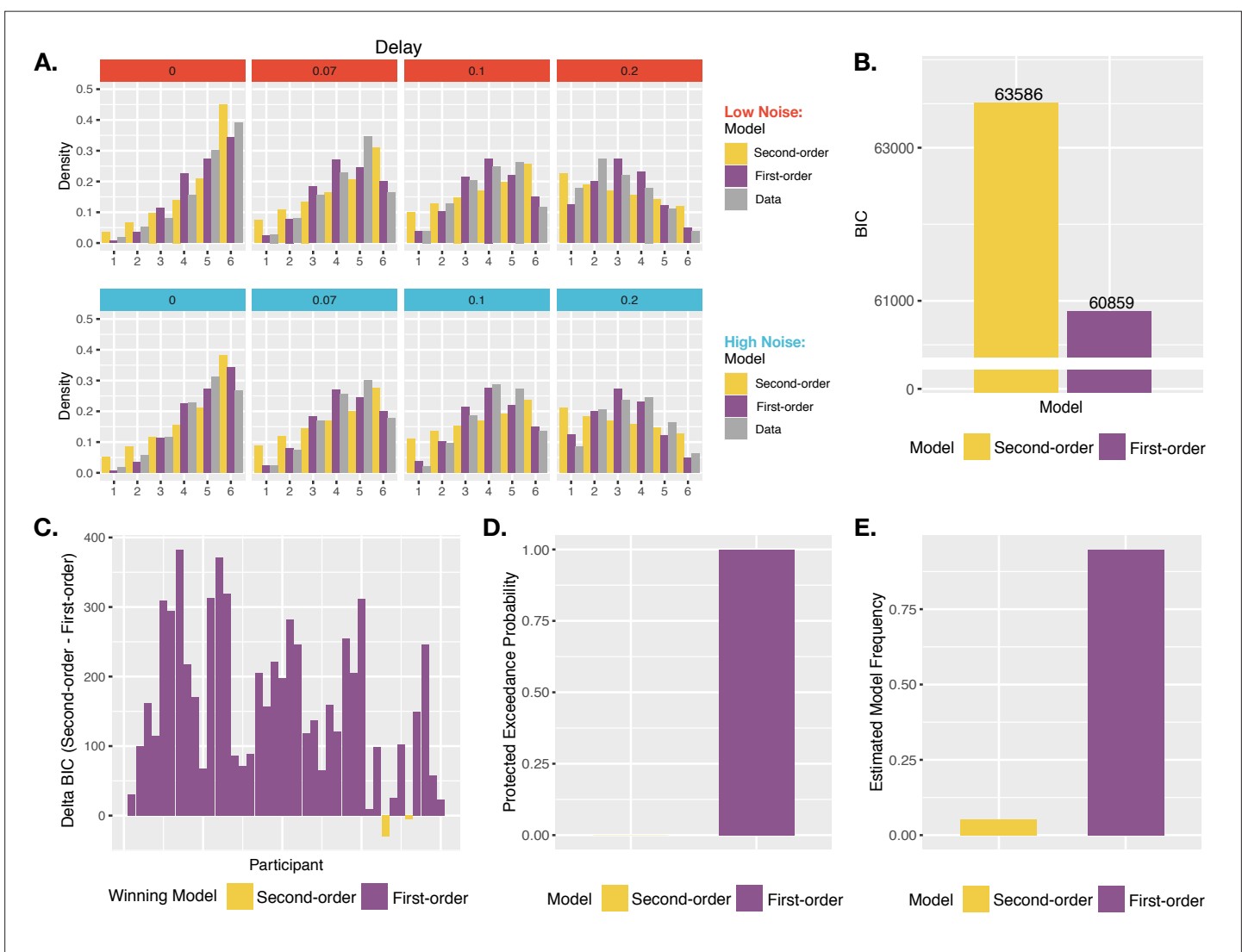

**Figure 4.** Agency model fits and results. (**A**) Model fits. Simulated probability of each JoA for a given delay and noise level, given best fitting parameters from the MLE analysis for Second-order and First-order models. These are portrayed as densities across the JoAs. Data distributions are shown in shaded gray, also portrayed as densities across discrete ratings. (**B**) Group BIC Results. BIC comparison between Second-order and First-order models on pooled JoA data. (**C**) Subject-wise BIC Results. Difference in BIC (Second-order - First-order) for each participant, with negative values indicating that the Second-order model fit better, and positive values indicating that the First-order model fit better. (**D**) Protected Exceedance Probabilities. Protected exceedance probabilities of the Second-order and First-order models. (**E**) Estimated Model Frequencies. Predicted model frequencies estimated from the exceedance probability analysis.

in terms of the actual strength of the agency experience. In contrast, in the Second-order model, a JoA of '6' would always reflect the same perceived level of certainty about the agency decision and hence the same agency experience, since JoAs combine the evidence strength and sensory noise.

We compared the two models in their ability to account for the distributions of JoAs per delay and noise level (*Figure 4A*) at the group level (with pooled data) and at the single-participant level. We also measured protected exceedance probabilities (PEPs) to assess the probability that each model was the most frequently occurring one in the population, adjusted for the ability to reject the null hypothesis (that the models are equally likely in the population). To compare the models, we first found the best fitting parameter values for each model — low noise $\sigma_L$, high noise $\sigma_H$, decision criterion (c), and the mapping parameter, or the number of ratings to be considered as 'Yes' responses (NRYes), for both models, as well as the scale range parameter (s) for the First-order model — using maximum likelihood estimation (MLE), and then performed a Bayesian-information criterion (BIC) comparison. Following standard recommendations (*Raftery, 1995*), we required a minimum BIC difference of 2 in order to consider one model a better explanation of the data than the other. To perform the Bayesian model selection and calculate the PEPs, we used the 'bmsR' package (*Lisi, 2021*) with model evidence computed from the Akaike weights (*Penny, 2012*; *Rigoux et al., 2014*; *Stephan et al., 2009*; *Wagenmakers and Farrell, 2004*). We also performed a model recovery analysis across the set of winning parameter values of each model across participants, confirming that the models were distinguishable from one another over the entire relevant parameter space (see Appendix 4).

Against the notion that agency ratings and confidence arise from analogous uncertainty monitoring computations, the group-level analysis revealed that the First-order model could better explain the JoA data ($\Delta BIC_{Second-First}$ = 2728, *Figure 4B*). The best fitting parameters for this First-order model were $\sigma_L$ = 0.16, $\sigma_H$ = 0.16, (c) = 0.16, NRYes = 3, and (s) = 1.11. In comparison, the best fitting parameters for the Second-order model were $\sigma_L$ = 0.19, $\sigma_H$ = 0.24, (c) = 0.16, and NRYes = 3. The predicted densities of each rating per delay and noise level for each model's best fitting parameters can be seen in *Figure 4A*. Fitting the two models to each participant revealed results consistent with the group-level analysis: The First-order model could better explain the data for 38 out of 40 participants, whereas the Second-order model provided a better fit for only 2 (*Figure 4C*). Also in line with this, the PEPs indicated that the First-order model occurs most frequently in the population (*Figure 4D*), with the predicted frequencies shown in *Figure 4E*. This was despite the First-order model including one additional free parameter, compared to the Second-order model.

We then performed the same model comparison on confidence ratings from the confidence task in order to confirm metacognitive computations underlying confidence using this modeling approach, for comparison to our agency-rating results. As expected, the Second-order model could better explain confidence ratings ($\Delta BIC_{First-Second}$ = 1121, *Figure 5A*), suggesting confidence to involve metacognitive computations, in contrast to JoAs. The PEPs also suggested the Second-order model to be the most frequently occurring model in the population (*Figure 5C*), with estimated frequencies shown in *Figure 5D*. The subject-wise BIC comparison revealed the Second-order model to provide a better fit in 24 out of 40 participants, the First-order model to provide a better fit in 13, and a BIC difference of less than 2 (suggesting neither model being a conclusively better fit) in 3 (*Figure 5B*).

## Discussion

Previous research has investigated the effect of noise on implicit proxies of agency such as temporal binding effects, but it has remained unclear how explicit agency judgments incorporate uncertainty. It is possible that JoAs adopt the effect of uncertainty at the first-order level, by taking a direct readout of the variable lower level signal. However, it is also possible that JoA computations involve a second-order estimate of the noise of first-order signals, which we would take to be a metacognitive computation. Here, we tested whether JoAs are metacognitive in a computational sense by examining whether they involve the same second-order uncertainty monitoring computations as metacognitive confidence ratings, or whether they can only be said to be metacognitive at a broad conceptual level. We found that uncertainty did not affect agency ratings in the same way as it does confidence ratings, suggesting that any effect of noise is better considered as an influence on first-order signals, without involving metacognitive noise estimates.

By combining two tasks, we brought JoAs into a standard metacognitive framework and compared them to confidence ratings following a 2IFC decision. We examined how discrimination accuracy

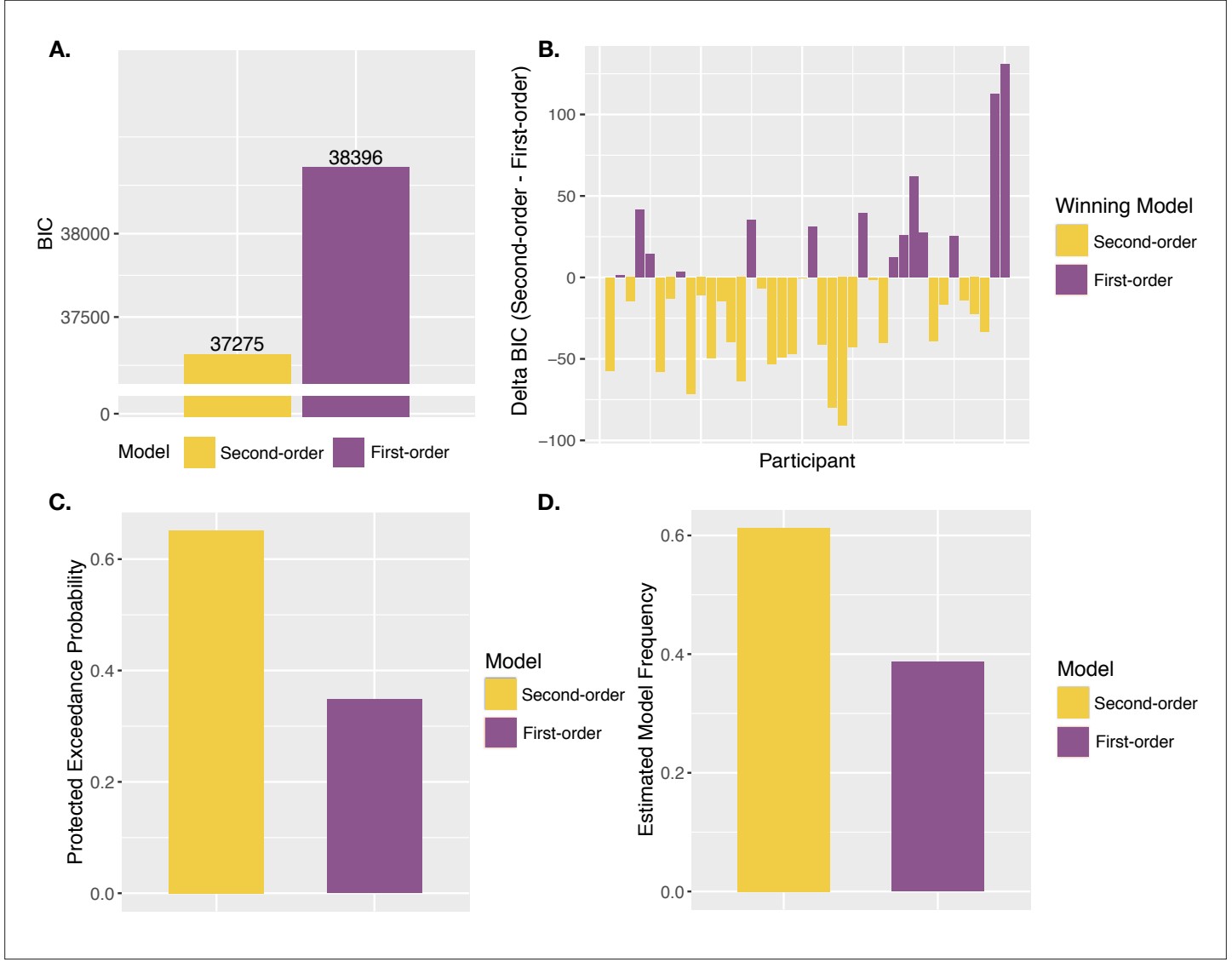

**Figure 5.** Confidence model results. (**A**) Group BIC Results. BIC comparison between the Second-order and First-order models on pooled confidence rating data. (**B**) Subject-wise BIC Results. Difference in BIC (Second-order - First-order) for each participant, with negative values indicating that the Second-order model fit better, and positive values indicating that the First-order model fit better. (**C**) Protected Exceedance Probabilities. Protected exceedance probabilities of the Second-order and First-order models. (**D**) Estimated Model Frequencies. Predicted model frequencies estimated from the exceedance probability analysis.

and confidence changed with sensory noise in the confidence-rating task, to first confirm that the sensory noise manipulation had the intended effect. Observing the expected effects of sensory noise on confidence allowed us to consider whether JoAs in the agency-rating task responded to sensory noise in a computationally analogous way. We reasoned that, if this were the case, JoAs would satisfy two criteria: First, they would depend on the precision of the comparator information, reflecting more than just a readout of the perceived signal. Second, this dependence on noise would reflect underlying metacognitive computations such as those involved in confidence, in particular, second-order estimates of one's own sensory noise. The JoAs satisfied the first of the test-criteria. We found that noise did indeed influence mean JoA across delays, indicating that the noise condition is factored into JoAs. However, because the noise manipulation changed the display in a visually obvious way, this information about the condition could have influenced judgments in a way that did not reflect participants making metacognitive estimates of the noise of their own processing. The second test-criterion investigated precisely this possibility.

To assess the second criterion, we compared two computational models. As a prerequisite, both models satisfied the first test-criterion. We contrasted a Second-order agency model, which included metacognitive noise estimates, with a First-order model that did not imply metacognitive processing. We tested these models in their ability to fit participants' JoAs as well as confidence judgments, to understand the computations underlying both. In the case of confidence, the model comparison revealed that participants' judgments were best explained by the Second-order model, confirming the metacognitive computations underlying them (*Kepecs and Mainen, 2012*; *Meyniel et al., 2015*; *Pouget et al., 2016*; *Sanders et al., 2016*). In striking contrast, this model comparison against participants' agency ratings revealed that JoAs were better explained by the First-order model as compared to the Second-order agency-model. The First-order model accounted for the observed behavioral relationship between JoA and noise by assuming that participants compared trials only to other trials within the same condition, and set condition-specific maxima of their rating scales accordingly. It would be an interesting direction for future work to test how JoAs depend on sensory noise under a noise manipulation that is not easily detectable, to investigate if the behavioral relationship we observe between noise and JoAs is limited to cases in which participants can treat the manipulation conditions as independent contexts, as our models suggest.

Taken together, our results suggest that while JoAs can be influenced by sensory noise, this influence is not indicative of metacognitive processing, and JoAs may better reflect first-order assessments of agency signals. We therefore argue that greater care should be taken when discussing agency within a metacognitive context, as the assumptions made about agency judgments being metacognitive do not hold on a computational level. Although this work used confidence as a benchmark for metacognitive processing, the computation of interest is second-order monitoring of the precision of one's processing, which has become the narrower focus of recent metacognition work (*Fleming and Lau, 2014*). While JoAs may still satisfy broad definitions of metacognition, our results suggest that they may not satisfy this narrower definition that is associated with a concrete computational view. In this sense, these results may help specify and clarify the assumed relationships between explicit agency judgments and metacognition. At the same time, they may add to our understanding of how JoAs respond to uncertainty — similarly to first-order perceptual judgments — which is critical for interpreting agency reports, especially in certain clinical cases involving agency disruptions.

Importantly, we also found that participants can make metacognitive confidence judgments *about* agency decisions, but that subjective agency ratings do not share these computations, despite the same basic agency task and noise manipulation. The use of 2IFC agency tasks with confidence has recently been proposed as a promising step toward a more reliable and complete investigation of agency processing, both in healthy and clinical populations (*Wang et al., 2020*). Here, using a virtual hand, we extended this approach into a more proximal form of embodied agency, closer to agency over the body itself (*Christensen and Grünbaum, 2018*; *Dogge et al., 2019*; *Stern et al., 2020*; *Wen, 2019*), and provided an initial step in demonstrating that participants can meaningfully monitor the accuracy of these agency decisions. We suggest that confidence judgments about agency should be considered as the metacognitive level of an agency processing hierarchy, with agency judgments as explicit first-order judgments. This also brings agency in line with recent motor metacognition research that considers agency-like judgments such as decisions of which trajectory was caused by one's movement to be the first-order motor judgments, followed by metacognitive confidence ratings (*Arbuzova et al., 2021*).

Although we suggest that JoAs do not imply metacognitive uncertainty monitoring, the dependence of agency ratings on the noise condition should not be overlooked and may be highly relevant both for future experimental design and in the interpretation of explicit agency reports. This finding is in line with multifactorial accounts of JoAs as involving a variety of both internal and external cues (*Synofzik et al., 2008*), and with the expanding empirical work investigating a range of contextual effects on agency (*Minohara et al., 2016*; *Wen, 2019*). Our work also fits within cue integration theories of agency (*Moore and Fletcher, 2012*; *Synofzik et al., 2013*), with the delay information being weighted less heavily when the signal is made less precise. Further, these results are relevant to empirical work examining cue integration in agency, as they suggest that having a perceivable manipulation such as reduced visibility in order to add noise to feedback cues may itself act as an additional factor influencing agency judgments, which should be accounted for in design and analysis.

These findings also complement recent work that has aimed to find computational models of agency, but has focussed on low-level FoA and implicit measures such as temporal binding effects (*Legaspi and Toyoizumi, 2019*). Here, we bring explicit agency judgments into a Bayesian and SDT framework, implementing formal computational models that could be used to further assess computations underlying different JoAs. Our findings support the suggestions of previous work that, while Bayesian confidence computations may underlie pre-reflective FoA, explicit JoAs reflect a different computational mechanism, and factor in different contextual information (*Legaspi and Toyoizumi, 2019*; *Wen, 2019*).

Taken into the context of two-step models of agency (*Synofzik et al., 2008*), our results suggest that sensory noise information may influence lower level, perceptual agency signals by making them more variable, and that these more variable signals then feed into higher level JoAs, but that estimates of the noise do not contribute as an additional cue to inform the JoAs. Hence, higher order agency states remain naive with regard to how noisy the lower level signals are. This could illuminate agency processing deficits in clinical cases which might involve perceptual agency cues becoming very noisy due to low level processing deficits. If explicit JoAs do not consider this uncertainty (as our findings indicate), this could lead to extreme agency reports despite unreliable evidence and inaccurate agency inferences. This is in line with work on agency misattributions in schizophrenia suggesting that they may be due to particularly noisy low level agency information, possibly due to dysregulated neurotransmitter activity (*Corlett et al., 2010*; *Fletcher and Frith, 2009*; *Moore and Fletcher, 2012*; *Robinson et al., 2016*).

By combining an agency rating task with an agency discrimination and confidence task, we were able to consider JoAs against the benchmark of metacognitive confidence judgments, while keeping the basic stimuli and noise manipulations the same in both cases. However, the tasks still differed on the nature of the ratings: Ratings in the agency task were about the feeling of agency over a single movement, whereas ratings in the confidence task were about the accuracy of the decision in the 2IFC task. Importantly, we note that the findings that we describe here do not rely on a comparison between tasks. The confidence task merely served as a positive control task to ensure that our noise manipulation had the expected effect on a metacognitive judgment and that the second-order model we built best predicted the results of computations we knew to be higher-order.

## Limitations

Our conclusions about the computations underlying the JoAs are limited by the manipulations used in the task. First, we manipulated noise externally, not directly internally, and only added noise to the perceived delay signals, rather than motor or somatosensory cues. We additionally did not manipulate action selection noise, which has been shown to influence agency independently of the comparator signal (*Wenke et al., 2010*). More research will be necessary to understand whether our conclusions also apply to these other sources of noise in agency processing.

Further, our conclusions hold for the agency manipulation we used, namely altering the timing of the action outcome. But, it is unclear whether these findings apply to other kinds of agency manipulations. This approach is prevalent in the literature, so our findings are directly applicable to agency as it is commonly measured in experimental situations (*Blakemore et al., 1999*; *Farrer et al., 2013*; *Stern et al., 2021*; *Wen et al., 2015b*). More conceptually, the efficacy of delay manipulations has been argued to be valid for the study of more proximal agency over the body, and in external agency cases in which the timing of the outcome is well-established with precise expectations, but has been called into question as a device to truly affect sense of agency over the effects of our actions on the environment (*Wen, 2019*). Our virtual hand manipulation allows us to get close to this proximal body agency while still using a delay manipulation, and, even if viewed as an external outcome, the timing of the virtual hand movement is tightly linked to our precise expectations about the timing of our own movements (*Krugwasser et al., 2019*; *Stern et al., 2020*; *Stern et al., 2021*). Previous work using this virtual hand agency paradigm has also validated the approach of introducing delay in the visual feedback against other measures of agency (*Krugwasser et al., 2019*; *Stern et al., 2020*).

Our results should also be considered in the context of the particular agency rating scale we used, as this will constrain participants' rating behavior. Similar agency scales are commonly used in agency research (*Dewey and Knoblich, 2014*; *Imaizumi and Tanno, 2019*; *Kawabe et al., 2013*; *Metcalfe and Greene, 2007*; *Miele et al., 2011*; *Sato and Yasuda, 2005*; *Stern et al., 2020*; *Voss et al.,*

*2017*; *Wen et al., 2015a*), so the results presented here are relevant to understand the computations of agency ratings discussed in the existing literature. Thus, our results suggest that the JoAs measured in the literature do not indicate metacognitive computations. On the basis of these results, we concur with recent work (*Wang et al., 2020*) that suggests that a 2IFC task on agency followed by a confidence judgment may be more adequate to measure a metacognitive component of agency processing. Future work could apply a similar modeling approach in order to understand whether other types of agency judgments, for example ones explicitly discriminating between self and other agents, show the same first-order behavior. Future work could also further investigate another simplifying assumption made here in our models, namely that trials in which, due to internal noise, participants experienced a negative delay, would be associated with strong evidence for agency. While this is a reasonable assumption in our task, where participants knew that the virtual hand tracked their hand movement and likely dismissed the possibility that the virtual hand moved prior to them, it may need to be adapted to fit other, more ecologically valid cases of agency processing.

## Conclusion

Here, we investigated whether the role of uncertainty in explicit JoAs is computationally metacognitive. The results suggest that JoAs can be influenced by sensory noise, but that this effect is best considered as a contextual cue that can impact participants' use of the rating scale (at least when detectable in the environment), rather than as the result of a second-order noise monitoring computation. We therefore suggest that JoAs best reflect first-order comparator signals, with the metacognitive level of agency processing being second-order confidence judgments about one's agency.

# Materials and methods

The experiment was pre-registered (osf.io/pyjhm), and we describe deviations from the pre-registered plan.

## Participants

We pre-registered a sample size of 32 participants (based on power estimates from similar tasks). We collected data until we reached 40 participants that displayed the basic and expected manipulation effect of illumination (see above). We tested 47 young, healthy participants between 18 and 35 years of age (M = 27.15, SD = 4.68) in Berlin. To participate in the study, we required that participants were right handed (Edinburgh Handedness Inventory score: M = 79.5, SD = 23.6), had no injury or condition preventing or restricting movement of the right index finger, had normal or corrected-to-normal vision, and were fluent in English. Subjects were compensated with eight euros per hour or with course credit and gave signed, informed consent before starting the experiment. The ethics committee of the Institute of Psychology at the Humboldt-Universität zu Berlin approved the study (Nr. 2020–29), which conformed to the Declaration of Helsinki.

## Setup

We used a LEAP Motion controller (Leap Motion Inc, San Francisco, CA) to track participants' hand motion and to control in real time the movement of a virtual hand displayed on the screen. The experiment ran on a Dell Latitude 5591 laptop (Intel core i5 with 16 GB of RAM) with a display resolution of 1920 × 1080 (refresh rate = 60 Hz) using software built in Unity 5.6.1, and was modified from software used in previous studies (*Krugwasser et al., 2019*; *Stern et al., 2020*). The computer was placed to the left of an opaque board, which occluded participants' right hand from view. Participants placed their right hand under the LEAP Motion tracker, which was fixed with its sensors facing downward. Blackout curtains were used during all testing to keep the lighting conditions within the room as consistent as possible.

## Procedure

The tasks we used built on a paradigm in which participants see on the screen a virtual hand that follows the movement of their own, with a given temporal lag (*Krugwasser et al., 2019*). This paradigm allowed us to examine a situation closer to the more embodied or 'narrow' sense of agency that relates to control of the body itself (*Christensen and Grünbaum, 2018*; *Dogge et al., 2019*;

*Stern et al., 2020*). Each participant completed two tasks: a confidence-rating task (*Figure 1B*) and an agency-rating task (*Figure 2A*). In both tasks, we manipulated sensory uncertainty by controlling the visibility of the virtual hand, which is computationally analogous to adding noise to the signal. This allowed us to compare the effect of noise on JoAs and confidence. Importantly, we manipulated sensory noise orthogonally to the decision variable, as done by previous work (*Bang and Fleming, 2018*; *de Gardelle et al., 2015*; *Spence et al., 2016*), which allowed us to precisely examine effects of noise without altering the true degree of control that the participants had over the virtual hand movement.

Prior to starting the experiment, participants completed the Edinburgh handedness scale. We then did a short thresholding procedure to set the illumination level that would be used in the high-noise condition of the main tasks. The baseline illumination levels were set in the Unity environment with a directional light intensity of 0 in the high-noise condition and 0.001 in the low-noise condition, but the brightness of the screen was then further thresholded in order to account for differences in eyesight and lighting conditions in the room. Participants placed their right hand on the table, under the LEAP tracker, and held it still. On each trial, participants first saw a fixation cross, followed by two consecutive presentations (separated by a flashed gray screen) of the virtual hand on the screen in the dark illumination condition, and in one case, it was artificially enlarged. Participants then discriminated which of the two intervals contained the larger hand. We ran this in blocks of 10 trials. If participants responded less than 70%, or more than 80% correctly in a block, or if they reported discomfort from straining to see the hand, we manually adjusted the brightness setting of the computer display. This rough thresholding procedure took approximately 5 min. The brightness was, however, further adjusted if participants reported not being able to see the virtual hand movement during the training or at the beginning of the task. The brightness was only re-adjusted prior to the confidence task for one participant.

## Agency-rating task

All participants performed the agency-rating task first so that subjective ratings would not be biased by the structure or ratings of the confidence task. Because the two tasks employed a very similar rating scale, we wanted to make sure that agency ratings corresponded to how they are collected in the literature and not that, for example, participants interpreted the agency scale as the confidence scale. This was important, as our main research question examined the computations underlying ratings in this task, whereas our confidence task served as a positive control for a known metacognitive judgment. On each trial, participants began with their right hand resting on the table, palm facing up, with fingers extended. They saw a fixation cross (for 1.5 s), then the virtual hand appeared and they had 2 s in which to flex and extend their index finger once. The virtual hand displayed their movement either in real time, or with an added delay of either 70, 100, or 200 ms. Participants then rated their agency over the virtual hand movement on a scale from 1 (lowest) to 6 (highest). We explained that the term 'agency' referred to how much control they had over the movement of the virtual hand, and they were asked to focus specifically on the timing of the movement. Agency ratings were made using arrow keys to move a cursor, which started at a random position on the six-point scale each trial. Additionally, error trials in which there was a glitch of the virtual hand (such as flipping or contorting), participants saw no virtual hand, or participants made the wrong hand movement, could be marked using the Space key. Overall, 2.4% of trials were marked as errors in the confidence task, and 1.9% were marked as errors in the agency rating task.

To achieve the high-noise condition, the virtual hand was displayed under dark, low-contrast illumination, using a directional light intensity of 0 in the Unity environment. In the low-noise condition, the virtual hand was displayed under brighter, higher contrast illumination using a directional light intensity of 0.001 (*Figure 1A*). The noise conditions as well as the four delay levels were counterbalanced and randomly distributed across each block. There were 60 trials per delay level and lighting condition, for a total of 480 trials, split across six blocks. Prior to this task, participants completed a short training consisting of 20 trials and including both noise conditions and all delays, but they never received any feedback regarding any task. The agency-rating task took approximately 45 min.

## Confidence-rating task

After the agency-rating task, participants did a confidence-rating task. Participants again flexed and extended their right index finger under the LEAP motion tracker, while looking at the virtual hand on the screen. In contrast to the agency-rating task, they made two consecutive movements, each cued by the appearance of the virtual hand, and separated with a blank gray screen. They then decided which virtual hand movement they had more agency over, and rated their confidence in their response from 1 (lowest) to 6 (highest). The difference in delay levels between the movements in each trial was staircased, with one of the two movements always having no delay and the other being adjusted according to an online 2-down-1-up staircasing procedure aiming to achieve an overall accuracy of approximately 71%. Only the low-noise condition was staircased, and the delays of the high-noise condition were set to match those of the low-noise. Participants made their decision and then confidence rating using the arrow keys, and an error trial could be marked during either the decision or confidence rating.

We manipulated noise in the same way as in the agency-rating task, and this was fully counterbalanced with which movement was the delayed one, with these factors randomly distributed across each block. The noise condition was always the same for both consecutive movements within a trial, so in this task, the sensory noise in the environment led to an internal comparison of the two noisy delay signals, and confidence was modeled as monitoring the noise of this discrimination. There were 100 trials per noise condition, for a total of 200 trials across five blocks. Prior to this task, participants completed another short training consisting of 10 trials, but only in the low-noise condition, to adjust to the new movement and response structure. The confidence task took approximately 45 min. At the end of the session, participants completed an informal debriefing.

## Analysis

We removed any trials marked as errors, and any trials with reaction times shorter than 100ms or longer than 8 s for any decision or rating.

We tested our main hypotheses using the 'lme4' package (*Bates et al., 2015*) in R (*R Development Core Team, 2020*) to build linear mixed-effects models. All models included by-participant random intercepts, and the model for the agency-rating task included random effects for the interaction of interest (*Table 1*). All hypotheses were tested using two-tailed tests and an alpha level of 0.05, and additionally using Bayes factors, which we computed with the 'BayesTestR' package (*Makowski et al., 2019*) using default priors. To compute Bayes factors for the logistic mixed-effects analyses, we built Bayesian models with the 'brms' package (*Bürkner, 2017*). For each of these Bayesian regressions, we ran 4 chains of 15,000 iterations, including 5000 burn-in samples for a total of 40,000 effective samples, and ensuring a R-hat close to 1. Effect sizes for results of the linear mixed-effects analyses were computed as $\eta^2_p$ using the 'effectsize' package (*Ben-Shachar et al., 2020*), with 95% confidence intervals reported when possible (large sample sizes resulted in some confidence intervals of width zero, and hence uninterpretable). The results of our linear mixed-effects analyses on confidence and JoAs were confirmed using ordinal models, built using the 'ordinal' package (*Christensen, 2019*).

To analyse metacognitive ability from the confidence task, we measured metacognitive efficiency (M-Ratio) using the HMeta-d' toolbox (*Fleming, 2017*). In this analysis, for the MCMC (Markov chain Monte Carlo method) procedure we used three chains of 15,000 iterations with an additional 5000 for adaptation, thinning of 3, and default initial values from JAGS (Just Another Gibbs Sampler). We also ensured that R-hat was approximately one for all sampling.

As a deviation from the pre-registered analyses, we included our second test-criterion and computational modeling analyses, and excluded instead some planned analysis of the variability of ratings, as, in hindsight, we reasoned that this would not help to clarify the link between agency and metacognition. To perform the Bayesian model selection and get the PEPs, we used the 'bmsR' package (*Lisi, 2021*), computing model evidence using Akaike weights from the MLE analysis and using $10^6$ samples.

## Modeling

To test whether JoAs reflect metacognitive computations, we compared two computational models which could both account for the observed effect of noise on agency ratings. Both are based on signal detection theory and a comparator model of agency, with the amount of delay between the real movement and virtual hand movement as the signal. However, under one model (the Second-order

model) agency ratings involve a second-order assessment of sensory noise in the same way that confidence judgments do; whereas under the other model (the First-order model), agency is based on only first-order estimates of the internal signal strength. Under the First-order model, the observed behavioral effect of noise results from participants scaling their ratings independently for each condition, without the need for metacognitive estimates of the noise level.

Under both models, JoAs result from a Yes/No decision of whether participants felt agency over the virtual hand movement, and this is scaled into a rating according to a function of the strength of the evidence. Modeling JoAs as involving this binary detection decision allowed us to examine whether agency ratings follow the computations involved in decision confidence, and is also in line with work treating agency as a binary judgment (*Fukushima et al., 2013*; *Spaniel et al., 2016*). We assume that participants set an internal decision criterion (c) which determines whether they detected a delay — thus judging a disruption in their agency —, or whether they detected no delay and therefore judged themselves to have agency over the virtual hand movement. Then, the different agency ratings are modeled as additional criteria on either side of (c). We model agency ratings as getting more extreme as the perceived signal (x) gets further from the decision criterion in either direction, or in other words, as evidence supporting the agency decision increases. This predicts that perceived delays that are very long relative to the decision criterion would lead to a 'No' decision with strong evidence, and in turn low JoAs, whereas perceived delays that are very short would lead to a 'Yes' decision with strong evidence, and high JoAs. Crucially, the two models differed in the function of internal evidence (*f*(x)) that determined the agency ratings, and in particular in the way this function was affected by sensory noise.

### Agency ratings in the Second-order model

The Second-order model assumed that agency ratings scale as a function of internal evidence in the same way as confidence, namely scaling with the posterior probability of being correct, given a choice and the internal signal (*Sanders et al., 2016*). Therefore, in this model, the agency rating is computed by estimating the probability that the agency detection was correct, given the perceived signal and detection decision. Because this probability computation depends on the level of sensory noise, the Second-order model predicts that noise will be factored into participants' JoAs.

In both models, we obtained the criterion values that split the continuous range of possible *f*(x) values into equidistant bins. For the Second-order model, because confidence reflects a probability, it is naturally bounded to 1. So for the Second-order model (with a 3:3 mapping, see below) this amounted to finding the criterion values that would lead to confidence levels of ⅓, ⅔, and 1. To estimate the positions of the criteria on the internal signal axis, we followed an analytical solution that defines confidence as

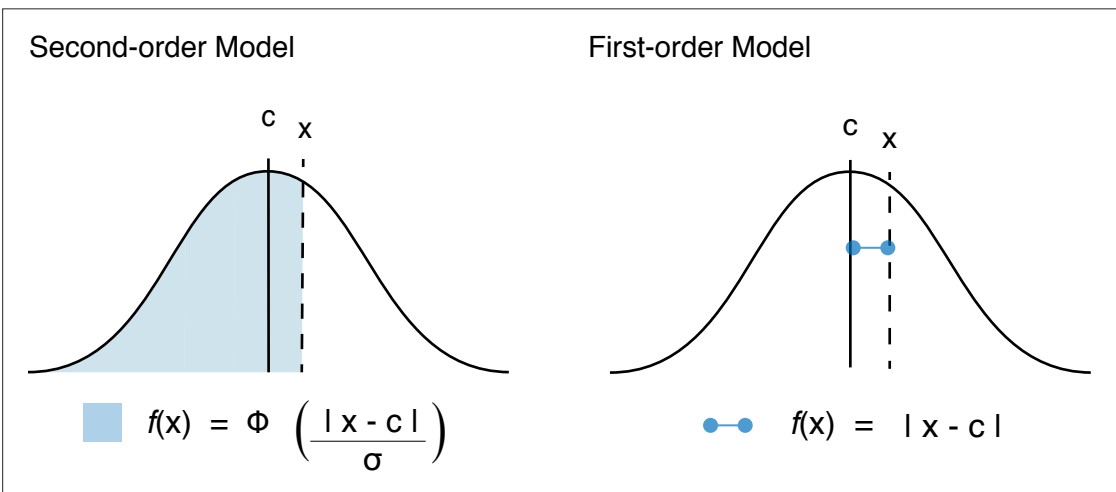

**Figure 6.** Function of internal evidence as estimated by each of the models tested. Agency as a function of evidence strength computations in each model. In the Second-order model this function of internal evidence reflects confidence based on the posterior probability of having given a correct response, given internal signal and choice. In the First-order model this is based on the perceived distance between signal and criterion.

$$\begin{cases} \Phi\left(\frac{c-x}{\sigma}\right) & if\ x \geq c \\ 1 - \Phi\left(\frac{c-x}{\sigma}\right) & if\ x < c \end{cases} \tag{1}$$

which we implemented, for convenience, as in a previous study (*Navajas et al., 2017*):

$$\Phi\left(\frac{|x-c|}{\sigma}\right) \tag{2}$$

This confidence measure can be interpreted as the perceived probability that the true delay signal was on the same side of the decision criterion as the internal signal, hence making the decision correct (*Figure 6*).

## Agency ratings in the First-order model

The non-Bayesian alternative model considers participants to compute their ratings proportionally to the distance between a point estimate of the internal signal and the decision criterion (*Figure 6*). According to this model, participants do not use the full distribution of internal signals in their assessment and hence do not make any metacognitive assessment of the precision of their evidence, but rather provide a rating that varies linearly as a function of the internal evidence, according to

$$f(x) = |x - c| \tag{3}$$

Unlike the Second-order model, ratings in this linear model are not inherently bounded at 1, as $f(x)$ could be as high as any arbitrarily high internal signal ($x$). Therefore, in order to find criterion placements that divide the continuous range of $f(x)$ values into equidistant bins, we needed to approximate a maximum. We assumed that participants bound their ratings based on the range of delays they experience throughout the experiment. Because we cannot know the true range of perceived delays, we approximated the most extreme perceived delays as the most extreme external signals plus a multiple of the noise, the freely fit *scale range* parameter ($s$). Based on the idea that participants rescaled their agency criteria according to the noise condition that they observed, accounting for the observed behavioral interaction effect, ($s$) acted as a multiple of the noise within a given condition. Hence, the scale range on low-noise trials would be from $-s\sigma_L$ to $[200+ s\sigma_L]$ but the scale range on high-noise trials would be from $-s\sigma_H$ to $[200+ s\sigma_H]$. However, critically, this noise parameter is not part of the internal JoA computation that we propose takes place in the brain, which does not involve estimates of the noise at all. It is only used by us, as experimenters, to guide an approximation of the range of perceived delays across the experiment, which we do not have external access to. Fitting this parameter allowed us to obtain the maximum $f(x)$ value as the maximum distance between ($c$) and either bound of the scale, and then divide the continuous $f(x)$ values into equidistant bins just as we did with the Second-order model. However, due to the different $f(x)$ computations, in this case the bins were equal linear distances on the internal signal axis, not equal probability bins as in the Second-order model (*Figure 3A*). Importantly, although the agency ratings do depend on the noise level due to the rescaling in this model, this does not involve participants making an assessment of the precision of their evidence, but just reflects participants considering each trial relative to a maximum that is different between illumination conditions. In other words, it would require less extreme evidence to lead to a '1' in the low-noise condition than a '1' in the high-noise condition.

Once we found criterion locations for each noise condition for each model, we calculated the probability of each rating for any given alteration and noise level. Using these probabilities, for all trials of a given participant or the pooled data, we built the likelihood function as

$$\prod_{\alpha=1}^{6} \left( \Phi\left(\frac{\gamma_{\alpha+1} - d}{\sigma}\right) - \Phi\left(\frac{\gamma_\alpha - d}{\sigma}\right) \right)^{n_\alpha} \tag{4}$$

where $\alpha$ indexes the agency rating criterion in a given noise condition; $\gamma_\alpha$ is the position of criterion $\alpha$, with $\gamma_1$ being $-\infty$ and $\gamma_7$ being $+\infty$; $d$ is the external delay; and $n_\alpha$ is the number of trials observed for that rating and delay, in that noise condition. We then took the product of this likelihood across all four possible external delays and across both noise conditions.

## Model parameters

Both models shared the parameters: standard deviation of the low-noise condition, $\sigma_L$, standard deviation of the high-noise condition, $\sigma_H$, and decision criterion (c). The First-order model also included the scale range parameter (s). Additionally, instead of assuming that participants always used half of the scale ratings to reflect detection of agency (JoA = 4:6), and half to reflect disruption to agency (JoA = 1:3), we fit a mapping parameter to capture the number of ratings used for each decision. We fit this parameter, NRYes, defined as the number of ratings used for 'Yes' decisions, with a minimum of one rating used for each decision. If NRYes is two, for example, this would suggest participants used ratings of '5' and '6' to indicate detections of agency, and ratings of '1' to '4' to indicate disturbances to their agency. By fitting this parameter, we avoided having to make any strong assumptions about how participants used the rating scale, considering we did not have their true Yes/No decisions.

## Modeling confidence

We also applied these two models to confidence ratings, in order to compare confidence computations with those underlying JoAs. For this analysis, the models were kept the same, except instead of fitting agency criteria, we fit the confidence criteria that divided confidence ratings into 12 total bins, with six ratings on each side of the decision criterion. We did not need to fit NRYes, as the assignment of ratings to a particular decision was forced by the task.

## Code availability

Reproducible analysis scripts and models are publicly available under https://gitlab.com/MarikaConstant/metaAgency, (copy archived at swh:1:rev:10a9c40ac4a8b45c81c5966297ef9911b2d33043, *Constant, 2022*).

## Acknowledgements

We thank Angeliki Charalampaki for discussions on the work presented here, and Matthias Guggenmos and Nathan Faivre for comments on an earlier version of this manuscript. MC was supported by the Deutsche Forschungsgemeinschaft (DFG, German Research Foundation) - 337619223/ RTG2386. MC and EF were supported by a Freigeist Fellowship to EF from the Volkswagen Foundation (grant number 91620). RS was supported by an Israeli Science Foundation Grant (ISF 1169/17). The funders had no role in the conceptualization, design, data collection, analysis, decision to publish, or preparation of the manuscript.

## Additional information

### Funding

| Funder | Grant reference number | Author |
| --- | --- | --- |
| Deutsche Forschungsgemeinschaft | 337619223 / RTG2386 | Marika Constant |
| Volkswagen Foundation | 91620 | Marika Constant Elisa Filevich |
| Israeli Science Foundation | ISF 1169/17 | Roy Salomon |

The funders had no role in study design, data collection and interpretation, or the decision to submit the work for publication.

### Author contributions

Marika Constant, Conceptualization, Data curation, Formal analysis, Investigation, Methodology, Software, Visualization, Writing – original draft, Writing – review and editing; Roy Salomon, Resources, Software, Writing – review and editing; Elisa Filevich, Conceptualization, Funding acquisition, Supervision, Writing – original draft, Writing – review and editing

## Author ORCIDs
Marika Constant http://orcid.org/0000-0003-3756-0362
Roy Salomon http://orcid.org/0000-0002-6688-617X
Elisa Filevich http://orcid.org/0000-0002-1158-8220

## Ethics
Human subjects: Subjects gave signed, informed consent before starting the experiment. The ethics committee of the Institute of Psychology at the Humboldt-Universität zu Berlin approved the study (Nr. 2020-29), which conformed to the Declaration of Helsinki.

## Decision letter and Author response
Decision letter https://doi.org/10.7554/eLife.72356.sa1
Author response https://doi.org/10.7554/eLife.72356.sa2

---

## Additional files

### Supplementary files
• Transparent reporting form

### Data availability
Raw data is publicly available under https://gitlab.com/MarikaConstant/metaAgency (copy archived at swh:1:rev:10a9c40ac4a8b45c81c5966297ef9911b2d33043).

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

## Appendix 1

Agency Rating Task Behavioural Results - All ParticipantsIn the analyses of the agency rating task that we report in the main text, we excluded data from participants for whom accuracy in the two conditions of the confidence task did not behave as expected by design. Specifically, if the noise manipulation that we used had the intended effect of making internal representations noisier, we expected accuracy in the high-noise condition to be lower than in the low-noise condition of the confidence task. However, we also ran the agency rating task analyses on the full dataset, without removing participants, and this had no significant impact on our conclusions. We ran the LME model on JoAs including the interaction between noise level and delay, and allowing for random effects per participant of the interaction, and random intercepts, without excluding any participants. As in what we report in the main text, we found a significant interaction effect between Noise and Delay, $F(1,56) = 37.53$, $p < 0.001$, $BF_{10} = 1.69 \times 10^4$, $\eta^2_p = 0.40$, 95% CI [0.21, 0.56], with a less extreme negative slope across delay values in the high-noise condition ($\beta_{High} = -5.64$, $SE = 0.65$), compared to low-noise ($\beta_{Low} = -8.91$, $SE = 0.79$), supporting the finding that JoAs met our first criterion. We also still found a significant main effect of Delay, $F(1,46) = 116.86$, $p < 0.001$, $BF_{10} = 250686$, $\eta^2_p = 0.72$, 95% CI [0.57, 0.80].

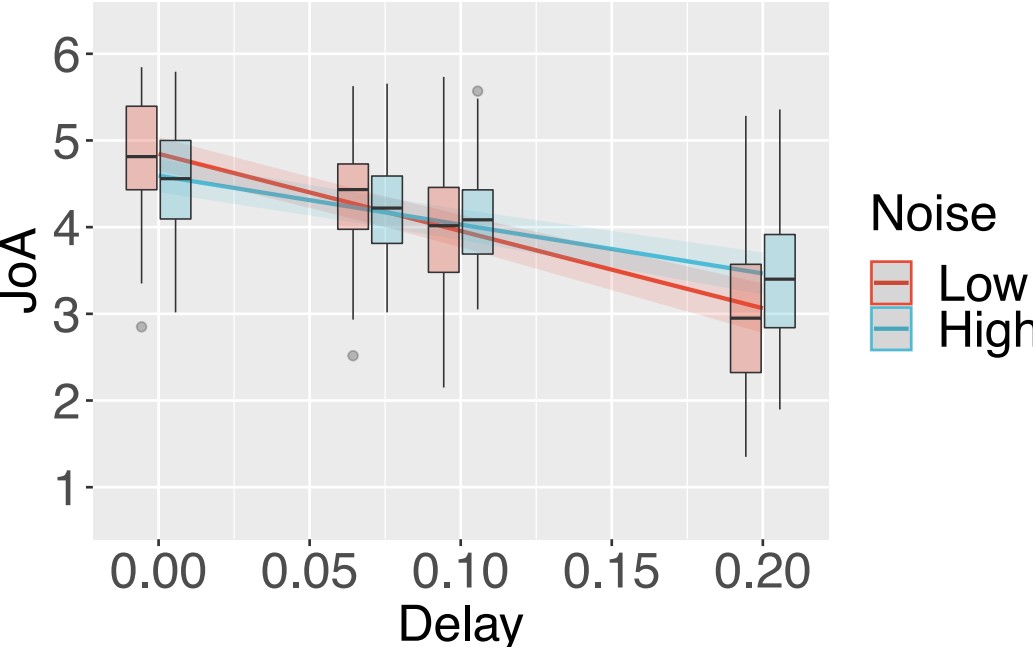

**Appendix 1—figure 1.** Interaction effect result with all participants. Predicted JoA across delays and noise conditions from linear mixed-effects model results. 95% confidence intervals shown. Boxplots reflect subjectwise mean JoAs per noise level and delay. They show the median, interquartile range (IQR) with hinges showing the first and third quartiles, and vertical whiskers stretching to most extreme data point within 1.5*IQR from the hinges. Outliers are plotted as gray dots. The model was run on data from all 47 participants.

## Appendix 2

### Confidence task regression analysis - ordinal models

Because participants rated confidence on a discrete scale from 1 to 6, we reran the linear mixed effects (LME) analysis from the Confidence Task using ordinal models. We built a model on confidence ratings including the interaction between response accuracy and noise level and each factor as fixed effects, response identity as an additional fixed effect, as well as by-participant random intercepts. This confirmed the results of our analysis using linear regression, and revealed a significant interaction between Response Accuracy and Noise, $\chi^2(1) = 14.86$, $p < 0.001$. This also confirmed the significant main effect of Response, $\chi^2(1) = 78.91$, $p < 0.001$.

### Agency rating task regression analysis - ordinal models

We also reran the LME analysis from the Agency Rating Task with ordinal models. We built a model on JoAs including the noise level and delay, and their interaction as fixed effects, as well as by-participant random effects of the interaction, and random intercepts. This confirmed the results of our LME analysis, revealing a significant interaction between Noise and Delay on JoAs, $\chi^2(1) = 43.15$, $p < 0.001$, as well as a significant main effect of Delay $\chi^2(1) = 23.56$, $p < 0.001$, replicating previous work (*Krugwasser et al., 2019*; *Stern et al., 2020*).

## Appendix 3

### Metacognitive ability

In order to further ensure that participants' confidence ratings in the Confidence Task reflected metacognitive processing, especially given relatively low accuracy under high noise, we analyzed their metacognitive ability in both noise conditions. We did this by computing metacognitive efficiency (M-Ratio), which accounts for differences in first-order task performance (*Maniscalco and Lau, 2012*), using the HMeta-d' toolbox (*Fleming, 2017*) for all participants. This revealed above-chance metacognitive efficiency (M-Ratio > 0) in both noise conditions. Importantly, we did not aim to compare metacognitive ability between conditions, but instead only confirmed that participants showed above-chance metacognitive performance in both conditions, suggesting in turn that confidence ratings were meaningful. Interestingly, we also found that metacognitive efficiency was nearly indistinguishable between conditions (M-Ratio$_{Low Noise}$ = 0.73, M-Ratio$_{High Noise}$ = 0.74).

### Measuring metacognitive sensitivity with logistic regressions

Along with our M-Ratio analysis, we also measured metacognitive sensitivity using logistic regressions, which does not account for first-order task performance. We built a mixed logistic regression model on Response Accuracy including Confidence and Noise as well as their interaction as fixed effects, and random Confidence slopes and random intercepts per participant (see below for model syntax). Metacognitive performance was estimated by considering the effect of confidence as a measure of how well confidence ratings tracked accuracy. We found a significant main effect of Confidence ($\chi^2(1) = 45.75$, $p < 0.001$, BF$_{10}$ = 1.24 $\times$ 10$^9$), which confirmed that participants' confidence ratings tracked their accuracy and were hence meaningful, despite the difficult task. Further, in line with the manipulation check displayed in *Figure 1C* (in the main text), we found a main effect of Noise on accuracy ($\chi^2(1) = 74.74$, $p < 0.001$, BF$_{10}$ = 4.14 $\times$ 10$^{12}$). The data are inconclusive on whether the two factors interact, as a frequentist analysis revealed a significant interaction between Confidence and Noise ($\chi^2(1) = 5.22$, $p = 0.022$), whereas Bayesian statistics revealed no conclusive evidence either way (BF$_{10}$ = 1.01).

### Model syntax

| Task | Hypothesis | Model formula |
|---|---|---|
| Confidence Task | Confidence tracks response accuracy, modulated by sensory noise | logit(Response Accuracy)~ Confidence*Noise + (Confidence \| Participant) |

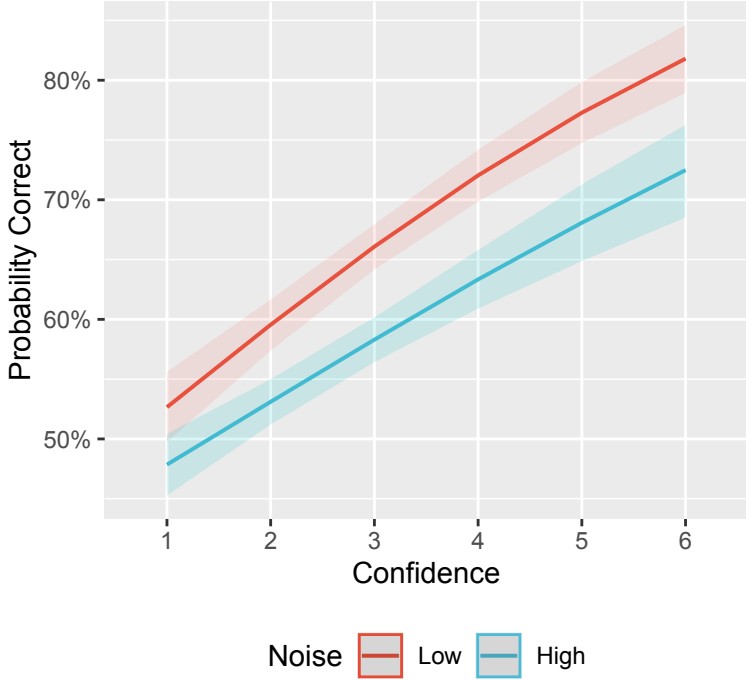

**Appendix 3—figure 1.** Logistic regression results. Metacognitive sensitivity quantified with predicted probability correct across confidence ratings and noise conditions from mixed logistic regression model results. We found a significant interaction between noise and confidence, with a smaller slope of confidence across accuracy under high noise. 95% confidence intervals shown.

## Appendix 4

### Model recovery analysis

To confirm that the Second-order and First-order models were distinguishable from one another, we ran a model recovery analysis. We took the winning parameters for each model for each participant and simulated 20 datasets with each parameter set. This generated 800 simulated datasets per model, covering all of the relevant parameters of our real participants. We then fit both models to those 1,600 simulated datasets and performed BIC comparisons as we did in our real data. We required a minimum BIC difference of 2 for one model to be considered a better fit, just as we did in our main modeling analyses. This revealed the Second-order model to be the better fitting model for 776 out of 800 of the datasets simulated with the Second-order model, suggesting it to be correctly recovered 97.00% of the time overall. Similarly, the First-order model was the better fitting model for 799 out of 800 of the datasets simulated with the First-order model, being correctly recovered 99.88% of the time overall. The mean BIC difference from models fit to 20 repetitions of each participant's relevant parameter set for each model are shown in the figure below. The mean ΔBIC (from the 20 repetitions) was above the minimum cutoff of 2, favoring the correct model, for all participants' parameter settings, for both models. Together, this analysis confirms the models to be distinguishable from one another across the entire relevant parameter space.

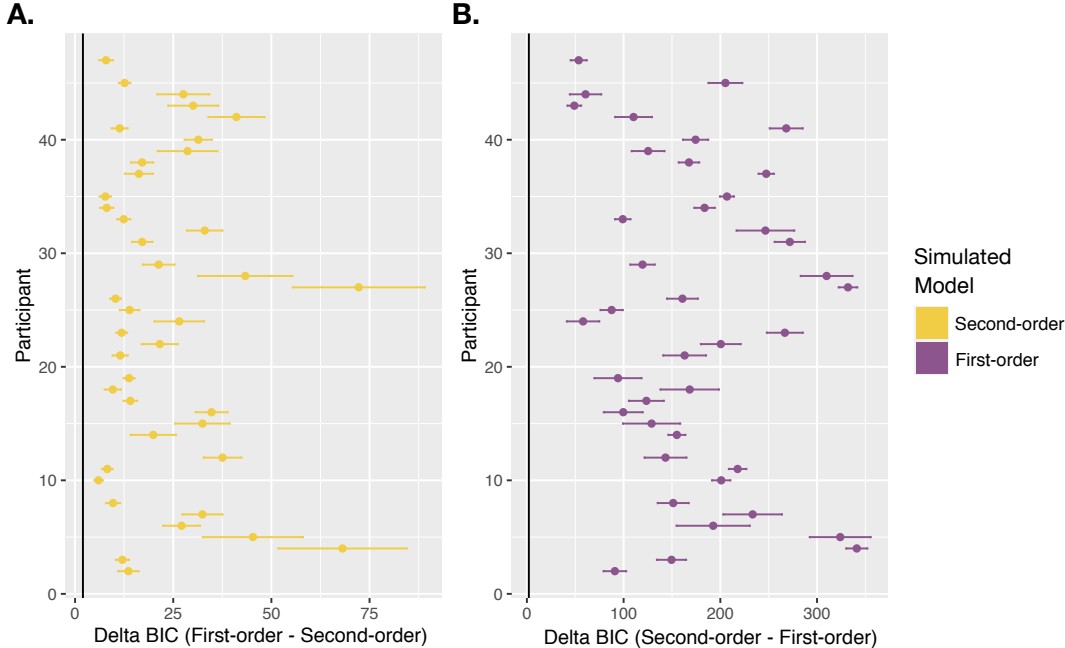

**Appendix 4—figure 1.** Model recovery results. BIC differences between the Second-order and First-order models, with positive values favoring the simulated model, fit to simulated datasets using all participants' winning parameter sets for each model. (**A**) BIC comparison results from datasets simulated with the Second-order model and the winning parameters for the Second-order model for each participant. (**B**) BIC comparison results from datasets simulated with the First-order model and the winning parameters for the First-order model for each participant. Points reflect the mean BIC difference across 20 repetitions of each parameter set, and error bars show the standard error. The black vertical line indicates the minimum BIC difference of 2 required for the correct model to be considered the better fitting model.

## Appendix 5

### Confidence model-fitting using additional information from confidence task

The confidence task provided us with additional information that could be used to fit the models, that we could not access in the agency rating task, namely, the calculated decision criteria and the difference in noise level between conditions, with $\sigma_H$ calculated as $\sigma_L * d'_H / d'_L$. Because we aimed to compare JoAs and confidence ratings in terms of their underlying computations, our main modeling analysis fit the two models to confidence ratings without using any of the additional information available, and subjecting the analysis to the same assumptions as with agency ratings. This involved freely fitting the noise levels, and assuming optimal decision criteria, as we did in the agency task. Here, we repeated the group-level analysis using the calculated decision criterion and noise difference. The calculated decision criteria for the pooled data were –0.079 in the low-noise condition and –0.13 in the high-noise condition. The high-noise level was calculated to be 2.73 times the low-noise, based on $d'_L/d'_H$. The winning parameters from this analysis for the First-order model were $\sigma_L = 0.73$, leaving $\sigma_H$ as 2.00, and (s) = 1.30. The winning parameter for the Second-order model was $\sigma_L = 0.37$, leaving $\sigma_H$ as 1.00. The Second-order model could still better explain confidence ratings ($\Delta BIC_{\text{First-Second}} = 1448$).

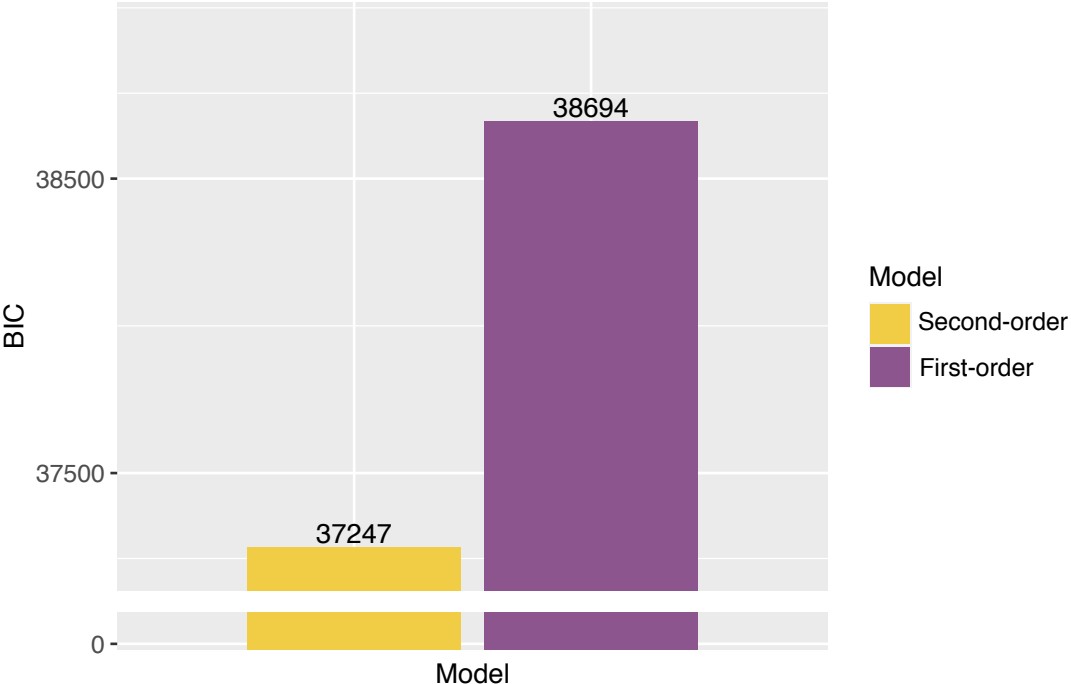

**Appendix 5—figure 1.** BIC results from confidence Task. BIC comparison between the Second-order and First-order models on pooled confidence rating data, using calculated decision criteria and difference between noise conditions.

