## [Editor Report]

This article describes a carefully designed study on the computational mechanisms underlying judgements of agency in an action-outcome delay task. Model-based analyses of behavior indicate that, unlike judgments of confidence, judgments of agency do not recruit metacognitive processes. This finding is important, because it challenges the assumed relation between agency and metacognition.

---

## [Decision Letter]

**Decision letter after peer review:**

Thank you for submitting your article "Judgments of agency are affected by sensory noise without recruiting metacognitive processing" for consideration by *eLife*. Your article has been reviewed by 3 peer reviewers, and the evaluation has been overseen by Valentin Wyart as the Reviewing Editor and Richard Ivry as the Senior Editor. The following individual involved in review of your submission has agreed to reveal their identity: Valerian Chambon (Reviewer #1).

As you will see below, all reviewers agree that the general approach you developed for your study of judgments of agency – combining experimental tasks with explicit computational modeling – is a clear strength of your study. The finding that sensory noise is monitored differently for judgments of agency and confidence judgments is interesting, and novel. However, there are different issues – in particular conceptual ones regarding the definition of metacognition but also experimental ones regarding the relation between the two tasks (and their associated judgments) you have used in your work. The essential revisions below have been discussed among reviewers and should be addressed in a point-by-point response. The individual reviews are also provided below for your information (please check all revisions regarding typos and concerns regarding the presentation of the results), but they do not require point-by-point responses. This is based on your responses to the essential revisions that we will evaluate your revised manuscript.

Essential revisions:

1) A first conceptual issue concerns the definition of metacognition in the study. It appears currently quite loosely defined, which is a problem for a study which makes specific claims about the distinction between metacognition and the sense of agency. In the present version of the manuscript, a metacognitive process appears to mean "being about first-order signals". But this loose definition is not satisfying when thinking about the rescaling model used to explain judgments of agency – which is seen as non-metacognitive. Indeed, this rescaling model still requires agents to track something about internal noise corrupting first-order signals, which would qualify it as metacognitive. A clear and unambiguous definition of metacognition should be provided upfront in the revised manuscript to avoid conceptual confusions.

For example, the authors argue their Bayesian model is a metacognitive one, because it requires the observer to have second-order access to an estimate of their own sensory noise. But even though the Bayesian model in this paper clearly incorporates an estimate of the noise/uncertainty in the signal, not all representations of noise are second-order or metacognitive. For example, Shea (2012) has noted that in precision-weighted Bayesian inference models throughout neuroscience (e.g., Bayesian cue combination, also discussed in this paper) the models contain noise estimates but the models are not metacognitive in nature. For example, when we combine a noisy visual estimate and a noisy auditory estimate, the Bayesian solution requires you account for the noise in the unimodal signals. But the precision parameters in these models do not necessarily refer to uncertainty in the agent's perceptions or beliefs, but uncertainty in the outside world. Similarly, in the Bayesian model proposed in this study, it is not clear why we should think of the uncertainty parameter as something metacognitive (e.g., about the agent's internal comparator representations) rather than something about the outside world (e.g., the sensory environment is noisy). This should be clarified and discussed in the revised manuscript.

Shea (2012) Reward prediction error signals are meta-representational. Nous, DOI: 10.1111/j.1468-0068.2012.00863.x

2) The present manuscript suggests that judgments of agency are subject to only one source of internal noise, the comparator. While it is understandable, given that the task is not likely to be associated with significant action selection (motor) noise, it is possible that this design choice has penalized the hypothesis of metacognitive monitoring of uncertainty by judgments of agency. This would require having a task design that manipulates selection difficulty (and thus selection noise), to see whether judgments of agency are sensitive to selection noise – which would potentially make them metacognitive (since judgments of agency would reflect second-order measures of this selection noise). It is indeed well known that selection noise affects judgments of agency (see, e.g., Wenke et al., 2010), independently of any comparison between predicted and observed signals. While we are not requiring you to perform additional experiments, this prior work and the limitations of the current study along these lines should be explicitly discussed in the revised manuscript.

Related to this issue, the validity of using an action-outcome delay task to generate broad conclusions about the nature of judgments of agency appears currently limited. In the task used in the study, the experience of agency depends *only* on interval detection, i.e., sensitivity to temporal contiguity. But while this is a popular approach in the field, the temporal contiguity actions and outcomes is only one of the cues that influence judgments of agency. Recent authors (e.g., Wen, 2020) have suggested that this manipulation may be problematic for a number of reasons. In similar types of paradigm, Wen (2020) notes that agents are able to accurately judge their control over action outcomes that are substantially delayed (e.g., well over 1000 ms) and thus it is possible that tasks manipulating action-outcome delays are biasing participants to report variance in the delays they experience rather than their actual experience about what they can and cannot control. Indeed, in the Methods section, the authors note participants were asked to "focus specifically on the timing of the movement" of the virtual hand, which may make this concern particularly salient. The judgment of agency made by participants can thus be reframed in this task as "did I detect a delay?", which limits the generalizability of the findings to many other situations where judgments of agency are not restricted to delay detection.

In practice, these concerns require an explicit discussion in the manuscript. You should at least consider explicitly whether your findings indicate that *sensorimotor delay judgements in particular* (rather than judgments of agency in general) are non-metacognitive. This alternative, more focused interpretation of the findings, is by no means uninteresting, but it has a somewhat narrower theoretical significance for the key debate used to frame the study ("do agency judgements monitor uncertainty in a metacognitive way?"). Arguments against this alternative (more specific) account should be provided in the discussion to support the interpretation that you have chosen to put forward (that judgments of agency in general are non-metacognitive). It is important that the title and framing of the paper remains as close as possible to what the findings of the study are.

Wen (2020). Does delay in feedback diminish sense of agency? A Review. Consciousness and Cognition, DOI: 10.1016/j.concog.2019.05.007

3) The relationship between the confidence task and the judgment of agency task is not entirely clear. Indeed, the confidence task measures confidence about a discrimination based on judgments of agency, whereas the judgement of agency task is about directly inferring agency from one stimulus. Therefore, the confidence task reflects by design a higher-order judgement about judgements of agency and, in this task setting, the judgments of agency appear to be treated experimentally as first-order judgements. It is unclear whether this choice of task design has triggered in itself the difference in computations underlying confidence and judgments of agency, and whether alternative task settings could show similar computations for the two types of judgments. This does not disqualify the main finding of the study, which is about determining which kind of computations underlie judgments of agency, but it is very important to discuss specifically the relation between the two judgments in this particular task, and how this relation may have something to do with the obtained findings. Further experimental work could – in principle – quash these worries – e.g., by manipulating agency in a different way but demonstrating the same effects of noise on confidence but not agency judgements. We are not requesting you to carry out these additional experiments, but they should be set as critical next steps for addressing the limitations of the current study.

*Reviewer #1 (Recommendations for the authors):*

1. I find the research question itself really interesting, but I wonder if the authors are arguing against a strawman ("agency judgements are often assumed to be metacognitive"). That agency judgements are assumed to be metacognitive is certainly true according to the references cited in the article (Metcalfe and Miele) but I am not sure that this is a widespread view in the field. To my knowledge, agency judgements are often described as high-level, post-hoc, reflexive or retrospective, but none of these qualifications imply that JoAs are metacognitive per se. More recent references suggesting that indeed JoAs are metacognitive might be needed here.

2. The discussion elaborates on what metacognition is (cognition about cognition, a process that involves 2nd order uncertainty monitoring computations, etc.) but I think a real definition of what a metacognitive process/representation/computation is would be needed in the introductory section, which lacks such a definition.

Is "being about" a first-order signal (whether that signal is perceptual, motor or memory-related) the minimum condition for something to be labelled "metacognitive"?

3. The task is a relatively simple motor task with little motor or premotor noise – in the sense that it does not specifically involve motor preparation or selecting a motor program from alternatives. This premotor/selection noise has been repeatedly shown (e.g. Wenke et al., 2010) to affect JoA, independent of any comparison between predicted and observed signals. Thus, according to this alternative hypothesis, the noise/uncertainty that feeds into participants' JoA does not come from a noisy comparator, as assumed in this paper, but comes directly from the action selection/preparation circuits – i.e. is due to competition between the selected and alternative action program and/or to blurred boundaries between alternative motor plans (e.g. Nachev 2005; Cisek, 2007)

For reasons of parsimony, which I can fully understand, the present study suggests that the JoA is subjected to only one source of internal noise, the comparator.

I wonder to what extent this choice penalizes the hypothesis of (metacognitive) monitoring of uncertainty by the JoA. Is it possible that participants' JoAs are more sensitive to internal selection noise than to comparator noise? This may require replicating the same task by manipulating the selection noise and measuring whether the agency reports reflect second-order measures of this selection noise. Note that I am not asking here for this experiment to be carried out, but perhaps the authors can comment on this.

4. A question of clarification: perhaps I missed something in the manuscript, but what is the internal noise that JoAs monitor? Logically, it should be the noise arising from the *comparison* between the first-order sensory signals (predicted and observed), a comparison that gives rise to the agentive experience itself. And if so, I am not sure I understand clearly what the source of the noise monitored by the confidence reports in the task is: is it the noise arising from the *comparison* between the two agentive experiences (which are themselves each the product of a noisy comparison between predicted and observed sensory signals)? Is it then reasonable to assume that this comparison, which gives rise to the confidence report, somehow inherits the noise from the *first-order* comparison that gives rise to the agentive experience?

*Reviewer #2 (Recommendations for the authors):*

I was a bit confused about the rationale behind the first criterion which JoA's have to meet in order to be considered metacognitve. It was unclear to me how the JoA's are hypothesized to be influenced by the sensory noise exactly, beyond just making them noisier? Is there a fundamental reason to expect agency ratings to increase or decrease in noisier conditions which we would expect a priori? Expecting 'an effect on agency ratings' sounds rather vague. The results show that the effect of the delay becomes smaller during high noise conditions, which makes a lot of conceptual sense. Maybe pre-empting this somehow before the results will make things a bit clearer?

The contrasting two models elegantly reflect different underlying psychological strategies and are very well designed and implemented. However, I feel like the explanations of the models in the main text are still relatively technical and potentially hard to understand for readers unfamiliar with these specific types of models. I think adding a few extra sentences per model explaining the models in more psychological terms would help (e.g. 'second-order access to estimate their own sensory noise' -> add something like 'i.e. be able to reflect on how noisy their own sensory processing is' and 'rescaling depending on the noise condition' -> 'e.g. give less extreme agency judgments under high noise conditions').

*Reviewer #3 (Recommendations for the authors):*

In line with these comments above, I would suggest that the authors amend the manuscript to make it clear how detecting action-outcome delays relates to agency detection mechanisms in general – ideally with a persuasive rebuttal of the kinds of concern that Wen (2020) provides. Without a strong reason to believe that action-outcome delay detection is directly measuring the agency detection process (which Wen, 2020 etc. give us cause to doubt), the generality of these conclusions seems potentially limited, and the broadbrush conclusions currently offered might need to be moderated accordingly.

At the same time, I think the authors should also be explicit about what they mean by a 'metacognitive computation'. The real novelty of their approach seems to be getting into the nitty gritty of what different computational models would predict. But if the authors agree with me that models can have uncertainty parameters without being metacognitive, then more needs to be done to justify why the Bayesian model is a metacognitive one. Of course, the authors may disagree with me, but a strong rebuttal of this concern and an explanation for why uncertainty parameters entail metacognition would be an important addition to the paper.

I have a few other points the authors might consider useful and which might help orient a reader:

1. The authors discuss in places that their tasks bring agency judgements into a "standard metacognition framework". But there are some important disanalogies. For example, in a perceptual metacognition task there is a clearly correct Type 1 answer (e.g., stimulus was present or absent) whereas the question posed in these tasks does here does not have an objectively correct answer. Regardless of the stimulus delay, the correct answer is always "I was the agent", so really the task is looking at variance in Type 1 and Type 2 judgements which is separate from the ground truth (i.e., they are always the agent on every trial). This strikes me as an important difference from the standard metacognition framework as it is applied to perception or memory judgements, and may thus be worth flagging explicitly to a reader.

2. The Gaussian schematics of the models in Figure 2, Figure 3 etc. are a bit opaque without describing what the underlying variable is. Making it clear these show a probabilistic representation of sensorimotor delays would make these more intelligible.

3. The authors split the paper into 'confirmatory' and 'exploratory' analyses. I understand why, given their pre-registration, but my personal feeling was this disrupted the flow of the paper somewhat, since it means the reader sees the confidence task, then the agency task, then a model of the agency data, then a return to the confidence task. Grouping the sections by task (e.g., confidence task/ confidence model/ agency task/ agency model) might build up the authors conclusions more naturally since it establishes 'what is metacognition like?' before then asking 'is agency like that?'. Of course, this is just a thought, and doesn't change the substance of what is presented.

4. In Figures that show model predictions or simulated data (e.g., Figure 3) I think it could be helpful to show simulated/predicted data in the same format as the original data displays (e.g., matching the plots of how data is shown in Figure 2c). This would make it easier for the reader to compare qualitative differences between the simulated and real data, and between each of the models. (Admittedly, this is done in Figure 4, but the granularity of the presentation is hard to translate back up to the big picture patterns observed in the experiments).

5. In standard use 'noise' is something added to stimuli to make discriminations harder. Here the signal strength is actually reduced by dimming the virtual lights, rather than adding a noise mask etc. Labelling the conditions as something like 'Signal strength – High/Low' (reversed accordingly) might be more appropriate.

6. p.22 "pre-reflexive FoA". I think the authors may mean 'pre-reflective'

---

## [Author Response]

Essential revisions:1) A first conceptual issue concerns the definition of metacognition in the study. It appears currently quite loosely defined, which is a problem for a study which makes specific claims about the distinction between metacognition and the sense of agency. In the present version of the manuscript, a metacognitive process appears to mean "being about first-order signals". But this loose definition is not satisfying when thinking about the rescaling model used to explain judgments of agency – which is seen as non-metacognitive. Indeed, this rescaling model still requires agents to track something about internal noise corrupting first-order signals, which would qualify it as metacognitive. A clear and unambiguous definition of metacognition should be provided upfront in the revised manuscript to avoid conceptual confusions.For example, the authors argue their Bayesian model is a metacognitive one, because it requires the observer to have second-order access to an estimate of their own sensory noise. But even though the Bayesian model in this paper clearly incorporates an estimate of the noise/uncertainty in the signal, not all representations of noise are second-order or metacognitive. For example, Shea (2012) has noted that in precision-weighted Bayesian inference models throughout neuroscience (e.g., Bayesian cue combination, also discussed in this paper) the models contain noise estimates but the models are not metacognitive in nature. For example, when we combine a noisy visual estimate and a noisy auditory estimate, the Bayesian solution requires you account for the noise in the unimodal signals. But the precision parameters in these models do not necessarily refer to uncertainty in the agent's perceptions or beliefs, but uncertainty in the outside world. Similarly, in the Bayesian model proposed in this study, it is not clear why we should think of the uncertainty parameter as something metacognitive (e.g., about the agent's internal comparator representations) rather than something about the outside world (e.g., the sensory environment is noisy). This should be clarified and discussed in the revised manuscript.Shea (2012) Reward prediction error signals are meta-representational. Nous, DOI: 10.1111/j.1468-0068.2012.00863.x

Thank you for addressing these important points. As we understand it, this comment raises four key questions: (1) what is our definition of metacognition, (2) how is our Bayesian model metacognitive, (3) how is our Rescaling model non-metacognitive, (4) does our Bayesian model involve the monitoring of truly internal signals.

1. What is our definition of metacognition?

We agree that clear definitions of metacognition and also agency are generally important and in the case of our manuscript, even more so. In the new version of the Introduction, we now mention explicitly that “…we commit to a definition of metacognition as a process that involves second-order uncertainty monitoring computations.”

We have also further clarified our definition of metacognition in the Introduction of the manuscript (pg. 3).

2. How is our Bayesian model metacognitive?

In line with the definition of metacognition that we commit to above, we completely agree with the reviewers that noise can have an influence that is not second-order. In fact, both of our models (Rescaling and Bayesian) include a role of noise in agency at the first-order level by adding variance to the delay signals, which we do not consider to be metacognitive. To call JoAs “metacognitive” we required them to exhibit evidence that they involve a second-order numerical estimate of the noise. The reviewers reference the work of Shea (2012). Thank you for pointing us towards this paper, it has helped us to further clarify how to best describe our models. In this, Shea gives a nice example of two contrasting roles of noise, one metacognitive and one not:

“Suppose I read 16°C on the slightly unreliable thermometer outside the window. If a friend then tells me it is 20°C outside, I will revise my estimate based on both sources of evidence, perhaps to an average of 18°C. If instead of telling me about the weather, she tells me about the thermometer—that it under-reads by 2°C—then I would also revise my estimate to 18°C. In the first case I am relying on my friend for some evidence she has about the world and forming a conclusion based on both sources of evidence. In the second case I am relying on her for some evidence about the accuracy of my first estimate, and revising that estimate accordingly.”

In the second case of this example, it is the second-order representation about the accuracy of the thermometer that makes it metacognitive. This mirrors the computation of our Bayesian model, assessing the perceived probability of being accurate in ones’ agency decision, and then scaling the JoA accordingly. However, we note that in Shea’s example the information about the thermometer focusses on bias rather than variance. Moving to the second point, our Bayesian model assumes that in forming JoAs, the brain takes the lower level agency signals, already impacted by noise, computes the perceived probability of the response being correct, and uses this to scale the agency rating. This use of the secondary representation of the precision of the agency signals is what fits it to our definition of a metacognitive process.

3. How is our Rescaling model non-metacognitive?

In contrast to the Bayesian model, the Rescaling model assumes that the brain takes the lower level agency signals, already impacted by noise, and just groups all the ‘light’ trials into six even bins and all the ‘dark’ trials into six even bins, in order to give ratings on the six-rating scale. This does not require any estimate of the accuracy or noise of the agency signals, but just relies on direct linear readouts of the perceived delays. The only reason a noise parameter factors into our implementation of the Rescaling model is to approximate the full range of perceived ‘light’ and ‘dark’ trials as experimenters, but the brain’s JoA computation proposed by this model requires no noise parameter.

4. Does our Bayesian model involve the monitoring of truly internal signals?

We argue that it is reasonable to assume that agency ratings result from the monitoring of noise in an internal signal for the following reasons:

First, building off a comparator model of agency, we assume that in our task agency signals are a result of the comparison between the external outcome feedback and the internal prediction representation, hence the agency signals first arise internally, at this comparison stage. While we realize that the noise is still being manipulated externally, our Bayesian agency model involves an assessment of the perceived probability of being correct based on the prediction error signal. So, the noise being monitored in the model is the noise at the stage of internal comparison to the prediction representation. Second, we used a noise manipulation that is orthogonal to the delay. With this design, the delay signal is itself not altered in the environment, with delay always being the same externally, but rather the mapping between the external delay signal and the internal representation is made noisier. We have clarified this point in the manuscript (pg. 5, 10 Figure 2 legend). Third, we followed the methods in other perceptual metacognitive literature that adds noise to the external environment in order to manipulate internal processing noise (Bang and Fleming, 2018; Gardelle and Mamassian, 2015; Spence et al., 2016). We have additionally added this external noise manipulation as a limitation, indicating that future work could investigate this research question using other ways of manipulating noise (pg. 23-24).

We have now renamed the models: We call the Bayesian model the Second-order model instead, to stress that the defining factor of this model is its second-order nature, and not its Bayesian computation. In contrast, we call the Rescaling model the First-order model. Additionally, we have clarified the distinction between the metacognitive Second-order model and non-metacognitive First-order model in both the model descriptions (pg. 34-35), Introduction (pg. 3) and Results (pg. 13-14). Note that we will continue to refer to the models by their original names in this response letter, but have changed them in the manuscript.

In short, this comment raised four important and connected questions. Briefly summarized, we answered these by arguing that (1) metacognition involves second-order noise monitoring, (2) the Bayesian – now called Second-order – model is metacognitive not by virtue of involving Bayesian computations but by virtue of relying on second-order estimates of the variance of first-order signals; whereas (3) the Rescaling now called First-order – model does not rely on these noise estimates. Finally, we argue that (4) the noise monitored in the Bayesian model is of the internal signal because agency judgments are made on an internal signal that depends on the internal prediction representations.

Bang, D., and Fleming, S. M. (2018). Distinct encoding of decision confidence in human medial prefrontal cortex. Proceedings of the National Academy of Sciences,115(23), 6082–6087. https://doi.org/10.1073/pnas.1800795115

Gardelle, V. de, and Mamassian, P. (2015). Weighting Mean and Variability during Confidence Judgments. PLOS ONE, 10(3), e0120870. https://doi.org/10.1371/journal.pone.0120870

Shea, N. (2012). Reward Prediction Error Signals are Meta-Representational. Nous (Detroit, Mich.), 48(2), 314–341.https://doi.org/10.1111/j.1468-0068.2012.00863.x

Spence, M. L., Dux, P. E., and Arnold, D. H. (2016). Computations underlying confidence in visual perception. Journal of Experimental Psychology: Human Perception and Performance, 42(5), 671–682. https://doi.org/10.1037/xhp0000179

2) The present manuscript suggests that judgments of agency are subject to only one source of internal noise, the comparator. While it is understandable, given that the task is not likely to be associated with significant action selection (motor) noise, it is possible that this design choice has penalized the hypothesis of metacognitive monitoring of uncertainty by judgments of agency. This would require having a task design that manipulates selection difficulty (and thus selection noise), to see whether judgments of agency are sensitive to selection noise – which would potentially make them metacognitive (since judgments of agency would reflect second-order measures of this selection noise). It is indeed well known that selection noise affects judgments of agency (see, e.g., Wenke et al., 2010), independently of any comparison between predicted and observed signals. While we are not requiring you to perform additional experiments, this prior work and the limitations of the current study along these lines should be explicitly discussed in the revised manuscript.

We agree that other sources of noise besides the feedback in the comparator are possible. We have now added a clarification (pg. 23) that our study does not include action selection, and that we did not manipulate ease or conflict, hence we cannot draw conclusions about the influence of selection noise on JoAs.

Related to this issue, the validity of using an action-outcome delay task to generate broad conclusions about the nature of judgments of agency appears currently limited. In the task used in the study, the experience of agency depends only on interval detection, i.e., sensitivity to temporal contiguity. But while this is a popular approach in the field, the temporal contiguity actions and outcomes is only one of the cues that influence judgments of agency. Recent authors (e.g., Wen, 2020) have suggested that this manipulation may be problematic for a number of reasons. In similar types of paradigm, Wen (2020) notes that agents are able to accurately judge their control over action outcomes that are substantially delayed (e.g., well over 1000 ms) and thus it is possible that tasks manipulating action-outcome delays are biasing participants to report variance in the delays they experience rather than their actual experience about what they can and cannot control. Indeed, in the Methods section, the authors note participants were asked to "focus specifically on the timing of the movement" of the virtual hand, which may make this concern particularly salient. The judgment of agency made by participants can thus be reframed in this task as "did I detect a delay?", which limits the generalizability of the findings to many other situations where judgments of agency are not restricted to delay detection.In practice, these concerns require an explicit discussion in the manuscript. You should at least consider explicitly whether your findings indicate that sensorimotor delay judgements in particular (rather than judgments of agency in general) are non-metacognitive. This alternative, more focused interpretation of the findings, is by no means uninteresting, but it has a somewhat narrower theoretical significance for the key debate used to frame the study ("do agency judgements monitor uncertainty in a metacognitive way?"). Arguments against this alternative (more specific) account should be provided in the discussion to support the interpretation that you have chosen to put forward (that judgments of agency in general are non-metacognitive). It is important that the title and framing of the paper remains as close as possible to what the findings of the study are.Wen (2020). Does delay in feedback diminish sense of agency? A Review. Consciousness and Cognition, DOI: 10.1016/j.concog.2019.05.007

This concern, regarding the generalizability of our findings to cases beyond delay manipulations, is certainly important. Part of our rationale for using an action-outcome delay task was its prevalent use in the field (Blakemore et al., 1999; Faivre et al., 2020; Farrer et al., 2013; Kalckert and Ehrsson, 2012; Krugwasser et al., 2019, 2021; Stern et al., 2020, 2021; Wen et al., 2015b). We are focussed on understanding whether JoAs, particularly as they are measured in the literature, involve metacognitive processing like they have been suggested to. Because of the dominance of this agency manipulation, we argue that our research question is central and applicable to a wide range of agency work.

With regard to whether we can generalize conclusions to judgments of agency more broadly, we understand and agree with the concerns raised by Wen (2019). However, there are some additional arguments that Wen makes that support the idea that our task effectively manipulates agency. First, it is suggested that delay is indeed an effective manipulation for disrupting agency over one's own body. With our use of the LEAP Motion Controller and virtual hand, and with the participants’ real hand obscured from view, our paradigm brought the action-outcome delay task into the realm of body agency. Though the virtual hand movement is still in the external world, this is similar to tasks such as those with video feedback of a rubber hand, discussed in this section of Wen’s review. Other work using the same LEAP paradigm as ours has also found clear correlations between different agency measures, including between this delay manipulation and spatial and anatomical alterations (Krugwasser et al., 2019; Stern et al., 2020). Further, even within situations of agency over the environment, Wen points out that delay will be more effective in cases with well-established and immediate expected outcomes, such as when turning on a light in contrast to pressing a button for an elevator. The timing of the virtual hand movement in our task is well-established and would involve a very narrow window of flexibility, hence we argue that delay should be effective at disrupting the feeling of agency.

We have added a discussion of this issue in the limitations section of the manuscript (pg. 24).

Blakemore, S. J., Frith, C. D., and Wolpert, D. M. (1999). Spatio-temporal prediction modulates the perception of self-produced stimuli. Journal of Cognitive Neuroscience, 11(5), 551–559. https://doi.org/10.1162/089892999563607

Faivre, N., Vuillaume, L., Bernasconi, F., Salomon, R., Blanke, O., and Cleeremans, A. (2020). Sensorimotor conflicts alter metacognitive and action monitoring. Cortex; a Journal Devoted to the Study of the Nervous System and Behavior, 124, 224–234. https://doi.org/10.1016/j.cortex.2019.12.001

Farrer, C., Valentin, G., and Hupé, J. M. (2013). The time windows of the sense of agency. Consciousness and Cognition, 22(4), 1431–1441. https://doi.org/10.1016/j.concog.2013.09.010

Kalckert, A., and Ehrsson, H. (2012). Moving a Rubber Hand that Feels Like Your Own: A Dissociation of Ownership and Agency. Frontiers in Human Neuroscience, 6, 40. https://doi.org/10.3389/fnhum.2012.00040

Krugwasser, A. R., Harel, E. V., and Salomon, R. (2019). The boundaries of the self: The sense of agency across different sensorimotor aspects. Journal of Vision, 19(4), 14–14. https://doi.org/10.1167/19.4.14

Krugwasser, A. R., Stern, Y., Faivre, N., Harel, E., and Salomon, R. (2021). Impaired Sense of Agency and Associated Confidence in Psychosis. https://doi.org/10.31234/osf.io/9wav7

Stern, Y., Ben-Yehuda, I., Koren, D., Zaidel, A., and Salomon, R. (2021). The Dynamic Boundaries of The Self: Serial Dependence In Embodied Sense of Agency. PsyArXiv. https://doi.org/10.31234/osf.io/fcv7t

Stern, Y., Koren, D., Moebus, R., Panishev, G., and Salomon, R. (2020). Assessing the Relationship between Sense of Agency, the Bodily-Self and Stress: Four Virtual-Reality Experiments in Healthy Individuals. Journal of Clinical Medicine, 9(9). https://doi.org/10.3390/jcm9092931

Wen, W., Yamashita, A., and Asama, H. (2015b). The influence of action-outcome delay and arousal on sense of agency and the intentional binding effect.Consciousness and Cognition, 36, 87–95. https://doi.org/10.1016/j.concog.2015.06.004

3) The relationship between the confidence task and the judgment of agency task is not entirely clear. Indeed, the confidence task measures confidence about a discrimination based on judgments of agency, whereas the judgement of agency task is about directly inferring agency from one stimulus. Therefore, the confidence task reflects by design a higher-order judgement about judgements of agency and, in this task setting, the judgments of agency appear to be treated experimentally as first-order judgements. It is unclear whether this choice of task design has triggered in itself the difference in computations underlying confidence and judgments of agency, and whether alternative task settings could show similar computations for the two types of judgments. This does not disqualify the main finding of the study, which is about determining which kind of computations underlie judgments of agency, but it is very important to discuss specifically the relation between the two judgments in this particular task, and how this relation may have something to do with the obtained findings. Further experimental work could – in principle – quash these worries – e.g., by manipulating agency in a different way but demonstrating the same effects of noise on confidence but not agency judgements. We are not requesting you to carry out these additional experiments, but they should be set as critical next steps for addressing the limitations of the current study.

The confidence task was indeed higher-order by design. This was intentional: we designed a task that would serve as a positive control for a metacognitive judgment. With this task, we made sure that the Bayesian model we built best predicted the results of computations we knew to be higher-order.

Importantly, the nature of the metacognitive control task does not affect our conclusions about the computations underlying agency judgements. These are purely based on the agency task alone, not on the comparison of the two tasks.

We have clarified this in the following sentences of the manuscript (pg. 23):

“However, the tasks still differed on the nature of the ratings: Ratings in the agency task were about the feeling of agency over a single movement, whereas ratings in the confidence task were about the accuracy of the decision in the 2IFC task. Importantly, we note that the findings that we describe here do not rely on a comparison between tasks. The confidence task merely served as a positive control task to ensure that our noise manipulation had the expected effect on a metacognitive judgment and that the second-order model we built best predicted the results of computations we knew to be higher-order.”

Separately, this point raises the concern that a different agency task structure might itself influence results. In other words, had we designed a different agency task, the conclusion that JoAs are not metacognitive may have been different. We agree with this possibility and include it as a limitation (pg. 24-25). However, our aim was to examine whether the types of agency judgments that are collected in the field involve metacognitive computations, as they are sometimes assumed to. For this reason, we chose to examine a form of agency judgment readily used in empirical work (Dewey and Knoblich, 2014; Imaizumi and Tanno, 2019; Kawabe et al., 2013; Metcalfe and Greene, 2007; Miele et al., 2011; Sato and Yasuda, 2005; Stern et al., 2020; Voss et al., 2017; Wen et al., 2015a). We agree that, when forced into certain higher-order task structures, metacognitive monitoring of agency signals does occur. In fact, this is demonstrated by our confidence task, in which we show participants’ ability to make metacognitive judgments about their agency decisions. This does not impact our conclusion that agency judgments in themselves, as they are often collected in the field, do not inherently include metacognitive processing. Future work could continue to examine this question using other agency tasks, which we now expand on as a prospective direction (pg. 25).

Dewey, J. A., and Knoblich, G. (2014). Do Implicit and Explicit Measures of the Sense of Agency Measure the Same Thing? PLOS ONE, 9(10), e110118. https://doi.org/10.1371/journal.pone.0110118

Imaizumi, S., and Tanno, Y. (2019). Intentional binding coincides with explicit sense of agency. Consciousness and Cognition, 67, 1–15. https://doi.org/10.1016/j.concog.2018.11.005

Kawabe, T., Roseboom, W., and Nishida, S. (2013). The sense of agency is action–effect causality perception based on cross-modal grouping. Proceedings of the Royal Society B: Biological Sciences, 280(1763), 20130991. https://doi.org/10.1098/rspb.2013.0991

Metcalfe, J., and Greene, M. J. (2007). Metacognition of agency. Journal of Experimental Psychology. General, 136(2), 184–199. https://doi.org/10.1037/0096-3445.136.2.184

Miele, D. B., Wager, T. D., Mitchell, J. P., and Metcalfe, J. (2011). Dissociating Neural Correlates of Action Monitoring and Metacognition of Agency. Journal of Cognitive Neuroscience, 23(11), 3620–3636. https://doi.org/10.1162/jocn_a_00052

Sato, A., and Yasuda, A. (2005). Illusion of sense of self-agency: Discrepancy between the predicted and actual sensory consequences of actions modulates the sense of self-agency, but not the sense of self-ownership. Cognition, 94(3), 241–255. https://doi.org/10.1016/j.cognition.2004.04.003

Stern, Y., Koren, D., Moebus, R., Panishev, G., and Salomon, R. (2020). Assessing the Relationship between Sense of Agency, the Bodily-Self and Stress: Four Virtual-Reality Experiments in Healthy Individuals. Journal of Clinical Medicine, 9(9). https://doi.org/10.3390/jcm9092931

Voss, M., Chambon, V., Wenke, D., Kühn, S., and Haggard, P. (2017). In and out of control: Brain mechanisms linking fluency of action selection to self-agency in patients with schizophrenia. Brain, 140(8), 2226–2239. https://doi.org/10.1093/brain/awx136

Wen, W., Yamashita, A., and Asama, H. (2015a). The Sense of Agency during Continuous Action: Performance Is More Important than Action-Feedback Association. PLOS ONE, 10(4), e0125226. https://doi.org/10.1371/journal.pone.0125226

Reviewer #1 (Recommendations for the authors):1. I find the research question itself really interesting, but I wonder if the authors are arguing against a strawman ("agency judgements are often assumed to be metacognitive"). That agency judgements are assumed to be metacognitive is certainly true according to the references cited in the article (Metcalfe and Miele) but I am not sure that this is a widespread view in the field. To my knowledge, agency judgements are often described as high-level, post-hoc, reflexive or retrospective, but none of these qualifications imply that JoAs are metacognitive per se. More recent references suggesting that indeed JoAs are metacognitive might be needed here.

We thank the reviewer for the encouraging feedback about our research question. We have now reframed the motivation behind our study (pg. 2-3). We hope that this framing maintains the justification for the research question of whether agency judgments are metacognitive without needing to motivate it purely based on this assumption in the literature. In addition, we have added more recent references to support that this assumption is still present in agency work (pg. 3).

With our new framing we highlight that, while previous studies have examined the role of uncertainty in lower level FoAs, it remains unclear how JoAs monitor uncertainty. We examine the two main possibilities for how this might occur: JoAs might simply adopt the influence of noise that already took place in lower level agency signals, or there might be secondary noise monitoring that occurs. We consider the second option to be metacognitive processing, and we clarify the reasoning for this (pg. 3).

2. The discussion elaborates on what metacognition is (cognition about cognition, a process that involves 2nd order uncertainty monitoring computations, etc.) but I think a real definition of what a metacognitive process/representation/computation is would be needed in the introductory section, which lacks such a definition.Is "being about" a first-order signal (whether that signal is perceptual, motor or memory-related) the minimum condition for something to be labelled "metacognitive"?

This has been addressed in response to the Essential Revisions (1) above.

3. The task is a relatively simple motor task with little motor or premotor noise – in the sense that it does not specifically involve motor preparation or selecting a motor program from alternatives. This premotor/selection noise has been repeatedly shown (e.g. Wenke et al., 2010) to affect JoA, independent of any comparison between predicted and observed signals. Thus, according to this alternative hypothesis, the noise/uncertainty that feeds into participants' JoA does not come from a noisy comparator, as assumed in this paper, but comes directly from the action selection/preparation circuits – i.e. is due to competition between the selected and alternative action program and/or to blurred boundaries between alternative motor plans (e.g. Nachev 2005; Cisek, 2007)For reasons of parsimony, which I can fully understand, the present study suggests that the JoA is subjected to only one source of internal noise, the comparator.I wonder to what extent this choice penalizes the hypothesis of (metacognitive) monitoring of uncertainty by the JoA. Is it possible that participants' JoAs are more sensitive to internal selection noise than to comparator noise? This may require replicating the same task by manipulating the selection noise and measuring whether the agency reports reflect second-order measures of this selection noise. Note that I am not asking here for this experiment to be carried out, but perhaps the authors can comment on this.

This has been addressed in response to the Essential Revisions (2) above.

4. A question of clarification: perhaps I missed something in the manuscript, but what is the internal noise that JoAs monitor? Logically, it should be the noise arising from the comparison between the first-order sensory signals (predicted and observed), a comparison that gives rise to the agentive experience itself. And if so, I am not sure I understand clearly what the source of the noise monitored by the confidence reports in the task is: is it the noise arising from the comparison between the two agentive experiences (which are themselves each the product of a noisy comparison between predicted and observed sensory signals)? Is it then reasonable to assume that this comparison, which gives rise to the confidence report, somehow inherits the noise from the first-order comparison that gives rise to the agentive experience?

This is a correct interpretation, we ask whether JoAs monitor internal noise arising from the comparison between the prediction and observed outcome. Then in the confidence task, we test whether confidence involves a monitoring of the noise of the comparison between these two comparator signals. This is in line with the assumptions of detection and discrimination confidence tasks, in which the former involve monitoring the noise of a single sensory stimulus, and the latter involves monitoring the noise of a comparison between two sensory stimuli. In the tasks used here, this lower level “sensory signal” is the comparator signal, which we capture as a prediction error signal. We have clarified this in the legend of Figure 2 and in the following sentence (Methods, page 30): “The noise condition was always the same for both consecutive movements within a trial, so in this task, the sensory noise in the environment led to an internal comparison of the two noisy delay signals, and confidence was modeled as monitoring the noise of this discrimination.”

Reviewer #2 (Recommendations for the authors):I was a bit confused about the rationale behind the first criterion which JoA's have to meet in order to be considered metacognitve. It was unclear to me how the JoA's are hypothesized to be influenced by the sensory noise exactly, beyond just making them noisier? Is there a fundamental reason to expect agency ratings to increase or decrease in noisier conditions which we would expect a priori? Expecting 'an effect on agency ratings' sounds rather vague. The results show that the effect of the delay becomes smaller during high noise conditions, which makes a lot of conceptual sense. Maybe pre-empting this somehow before the results will make things a bit clearer?

We have now clarified (Results, pg. 11) how precisely we would expect the sensory noise to influence JoAs behaviourally, beyond just making them noisier, and why we predict this a priori. We also appreciate the suggestion to pre-empt the result of high noise decreasing the effect of delay and have included this (Introduction, pg. 4): “We expected the effect of the delay to become smaller in high perceptual uncertainty trials, leading JoAs to be less extreme, as compared to trials with lower uncertainty.”

The contrasting two models elegantly reflect different underlying psychological strategies and are very well designed and implemented. However, I feel like the explanations of the models in the main text are still relatively technical and potentially hard to understand for readers unfamiliar with these specific types of models. I think adding a few extra sentences per model explaining the models in more psychological terms would help (e.g. 'second-order access to estimate their own sensory noise' -> add something like 'i.e. be able to reflect on how noisy their own sensory processing is' and 'rescaling depending on the noise condition' -> 'e.g. give less extreme agency judgments under high noise conditions').

We first thank the reviewer for the encouraging feedback about our modeling approach, and also for this good suggestion. We have now included more intuitive descriptions of the computations that each model includes (pg. 13-14).

Reviewer #3 (Recommendations for the authors):In line with these comments above, I would suggest that the authors amend the manuscript to make it clear how detecting action-outcome delays relates to agency detection mechanisms in general – ideally with a persuasive rebuttal of the kinds of concern that Wen (2020) provides. Without a strong reason to believe that action-outcome delay detection is directly measuring the agency detection process (which Wen, 2020 etc. give us cause to doubt), the generality of these conclusions seems potentially limited, and the broadbrush conclusions currently offered might need to be moderated accordingly.

We thank the reviewer for pointing us to the important issues discussed in the paper by Wen (2019). This has been addressed in response to the Essential Revisions (2) above.

At the same time, I think the authors should also be explicit about what they mean by a 'metacognitive computation'. The real novelty of their approach seems to be getting into the nitty gritty of what different computational models would predict. But if the authors agree with me that models can have uncertainty parameters without being metacognitive, then more needs to be done to justify why the Bayesian model is a metacognitive one. Of course, the authors may disagree with me, but a strong rebuttal of this concern and an explanation for why uncertainty parameters entail metacognition would be an important addition to the paper.

We certainly agree that models can have uncertainty parameters without entailing metacognition, and we have clarified and addressed this in response to the Essential Revisions (1) above.

I have a few other points the authors might consider useful and which might help orient a reader:1. The authors discuss in places that their tasks bring agency judgements into a "standard metacognition framework". But there are some important disanalogies. For example, in a perceptual metacognition task there is a clearly correct Type 1 answer (e.g., stimulus was present or absent) whereas the question posed in these tasks does here does not have an objectively correct answer. Regardless of the stimulus delay, the correct answer is always "I was the agent", so really the task is looking at variance in Type 1 and Type 2 judgements which is separate from the ground truth (i.e., they are always the agent on every trial). This strikes me as an important difference from the standard metacognition framework as it is applied to perception or memory judgements, and may thus be worth flagging explicitly to a reader.

We disagree with the reviewer that the confidence task is not analogous to a standard metacognitive task.

Here, we assume that each of the two intervals (or movements) in the 2IFC task lead to participants experiencing two different levels of control. This assumption is rooted in the literature on agency, where introducing a delay between the expected and observed effects of an action leads to decreased agency (Blakemore et al., 1999; Faivre et al., 2020; Farrer et al., 2013; Kalckert and Ehrsson, 2012; Krugwasser et al., 2019, 2021; Stern et al., 2020, 2021; Wen et al., 2015b). Therefore, there is in fact an objectively correct answer in the discrimination task: It is reasonable to assume that participants experienced more control over the movement presented in the interval with the shortest delay. In fact, this was one of the motivating factors behind using a 2IFC task instead of a detection task, in which participants would instead be asked if they were the agent. However, we agree that an interesting future direction would be to examine this in different agency tasks, for example one in which participants are asked to discriminate between two actions, one caused by them and one by a different agent. While less well-controlled, this paradigm might get closer to the experience of loss of agency. We now mention this in the limitations section of the manuscript.

Blakemore, S. J., Frith, C. D., and Wolpert, D. M. (1999). Spatio-temporal prediction modulates the perception of self-produced stimuli. Journal of Cognitive Neuroscience, 11(5), 551–559. https://doi.org/10.1162/089892999563607

Faivre, N., Vuillaume, L., Bernasconi, F., Salomon, R., Blanke, O., and Cleeremans, A. (2020). Sensorimotor conflicts alter metacognitive and action monitoring. Cortex; a Journal Devoted to the Study of the Nervous System and Behavior, 124, 224–234. https://doi.org/10.1016/j.cortex.2019.12.001

Farrer, C., Valentin, G., and Hupé, J. M. (2013). The time windows of the sense of agency. Consciousness and Cognition, 22(4), 1431–1441. https://doi.org/10.1016/j.concog.2013.09.010

Kalckert, A., and Ehrsson, H. (2012). Moving a Rubber Hand that Feels Like Your Own: A Dissociation of Ownership and Agency. Frontiers in Human Neuroscience, 6, 40. https://doi.org/10.3389/fnhum.2012.00040

Krugwasser, A. R., Harel, E. V., and Salomon, R. (2019). The boundaries of the self: The sense of agency across different sensorimotor aspects. Journal of Vision, 19(4), 14–14. https://doi.org/10.1167/19.4.14

Krugwasser, A. R., Stern, Y., Faivre, N., Harel, E., and Salomon, R. (2021). Impaired Sense of Agency and Associated Confidence in Psychosis. https://doi.org/10.31234/osf.io/9wav7

Stern, Y., Ben-Yehuda, I., Koren, D., Zaidel, A., and Salomon, R. (2021). The Dynamic Boundaries of The Self: Serial Dependence In Embodied Sense of Agency. PsyArXiv. https://doi.org/10.31234/osf.io/fcv7t

Stern, Y., Koren, D., Moebus, R., Panishev, G., and Salomon, R. (2020). Assessing the Relationship between Sense of Agency, the Bodily-Self and Stress: Four Virtual-Reality Experiments in Healthy Individuals. Journal of Clinical Medicine, 9(9). https://doi.org/10.3390/jcm9092931

Wen, W., Yamashita, A., and Asama, H. (2015b). The influence of action-outcome delay and arousal on sense of agency and the intentional binding effect. Consciousness and Cognition, 36, 87–95. https://doi.org/10.1016/j.concog.2015.06.004

2. The Gaussian schematics of the models in Figure 2, Figure 3 etc. are a bit opaque without describing what the underlying variable is. Making it clear these show a probabilistic representation of sensorimotor delays would make these more intelligible.

We have added an explicit description of these Gaussians as probability distributions of the internal comparator signal to each Figure legend.

3. The authors split the paper into 'confirmatory' and 'exploratory' analyses. I understand why, given their pre-registration, but my personal feeling was this disrupted the flow of the paper somewhat, since it means the reader sees the confidence task, then the agency task, then a model of the agency data, then a return to the confidence task. Grouping the sections by task (e.g., confidence task/ confidence model/ agency task/ agency model) might build up the authors conclusions more naturally since it establishes 'what is metacognition like?' before then asking 'is agency like that?'. Of course, this is just a thought, and doesn't change the substance of what is presented.

We agree that it might be intuitive to group the analyses by task. However, the logic of our analyses progresses from the behavioural analyses of the confidence task as a manipulation check, to the agency analyses for comparison. The agency analyses include both the behavioural and computational sections, as we assess our two criteria for JoAs being metacognitive. This was the flow that we wanted to capture. The confidence modeling at the end of the Exploratory Analyses was added as an additional validation of our modeling approach, to ensure that models give the predicted results for a known metacognitive judgment. Grouping by task (with the confidence task first) would involve describing this confidence modeling prior to the description of the agency models that we developed, which we think would make the paper difficult to follow. Alternatively, with the agency task described first, we would only be able to refer to the behavioural results of the confidence task after having presented the agency modeling, despite the confidence results serving as an important validation of our manipulation.

At the same time, we understand the reviewer’s point about the order of the results being confusing. To make the grouping into behavioural and then computational analyses clearer, we have now moved the Metacognitive Ability results to the description of the behavioural confidence task results under Confirmatory Analyses. However, we have also moved the full description of this Metacognitive Ability analysis to Appendix 3 due to word count constraints.

4. In Figures that show model predictions or simulated data (e.g., Figure 3) I think it could be helpful to show simulated/predicted data in the same format as the original data displays (e.g., matching the plots of how data is shown in Figure 2c). This would make it easier for the reader to compare qualitative differences between the simulated and real data, and between each of the models. (Admittedly, this is done in Figure 4, but the granularity of the presentation is hard to translate back up to the big picture patterns observed in the experiments).

We agree that this would be an intuitive way to capture the model predictions, and had indeed included Author response image 1 in an earlier version of our manuscript. However, we included these two specific models in our analyses because they were both able to account for the interaction effect we show in Figure 2c (and not, for example, a simpler model where agency ratings reflect the internal signal strength with no rescaling). Hence, by design, there are no qualitative differences between the models at this level of granularity. This is why we display the model predictions on the distribution of ratings in Figure 3 to highlight their differences, rather than to show that both could capture the observed behavioural effect.

**Author response image 1. sa2fig1:** 

5. In standard use 'noise' is something added to stimuli to make discriminations harder. Here the signal strength is actually reduced by dimming the virtual lights, rather than adding a noise mask etc. Labelling the conditions as something like 'Signal strength – High/Low' (reversed accordingly) might be more appropriate.

We note that, in our paradigm, the signal strength corresponds to the delay between the actual and virtual hand movements. Therefore, increasing signal strength would imply increasing the degree of delay. Because we intentionally use a manipulation that is orthogonal to the delay signal, we want to avoid referring to it as ‘Signal Strength’. We have instead labeled them as ‘Noise’ conditions to be consistent with other perceptual metacognition literature.

We realize that the literature presents various ways of manipulating discrimination performance, including adding noise and decreasing contrast or visibility. However, these are computationally analogous in their effect on d´. So, in our case, decreasing illumination is analogous to increasing noise. We have now added a sentence clarifying this reversed relationship to the Methods section of the manuscript (pg. 27).

6. p.22 "pre-reflexive FoA". I think the authors may mean 'pre-reflective'

Thank you for pointing this out, we have now fixed it.